# Sequence variant affects *GCSAML* splicing, mast cell specific proteins, and risk of urticaria

Ragnar P. Kristjansson[1,10], Gudjon R. Oskarsson [1,2,10], Astros Skuladottir [1], Asmundur Oddsson [1],
Solvi Rognvaldsson [1], Gardar Sveinbjornsson [1], Sigrun H. Lund [1], Brynjar O. Jensson [1],
Edda L. Styrmisdottir[1], Gisli H. Halldorsson [1], Egil Ferkingstad [1], Grimur Hjorleifsson Eldjarn[1],
Doruk Beyter [1], Snædis Kristmundsdottir [1,3], Kristinn Juliusson[1], Run Fridriksdottir[1], Gudny A. Arnadottir [1],
Hildigunnur Katrinardottir[1], Margret H. Snorradottir[1], Vinicius Tragante [1], Lilja Stefansdottir[1],
Erna V. Ivarsdottir [1,4], Gyda Bjornsdottir [1], Bjarni V. Halldorsson [1,3], Gudmar Thorleifsson [1],
Bjorn R. Ludviksson[2,5], Pall T. Onundarson[2,6], Saedis Saevarsdottir [1,2,7,8], Pall Melsted[1,4],
Gudmundur L. Norddahl[1], Unnur S. Bjornsdottir[8,9], Thorunn Olafsdottir [1,2], Daniel F. Gudbjartsson [1,4],
Unnur Thorsteinsdottir[1,2], Ingileif Jonsdottir [1,2], Patrick Sulem [1✉] & Kari Stefansson [1,2✉]

Urticaria is a skin disorder characterized by outbreaks of raised pruritic wheals. In order to identify sequence variants associated with urticaria, we performed a meta-analysis of genome-wide association studies for urticaria with a total of 40,694 cases and 1,230,001 controls from Iceland, the UK, Finland, and Japan. We also performed transcriptome- and proteome-wide analyses in Iceland and the UK. We found nine sequence variants at nine loci associating with urticaria. The variants are at genes participating in type 2 immune responses and/or mast cell biology (*CBLB*, *FCER1A*, *GCSAML*, *STAT6*, *TPSD1*, *ZFPM1*), the innate immunity (*C4*), and NF-κB signaling. The most significant association was observed for the splice-donor variant rs56043070[A] (hg38: chr1:247556467) in *GCSAML* (MAF = 6.6%, OR = 1.24 (95%CI: 1.20–1.28), *P*-value = 3.6 × 10$^{-44}$). We assessed the effects of the variants on transcripts, and levels of proteins relevant to urticaria pathophysiology. Our results emphasize the role of type 2 immune response and mast cell activation in the pathogenesis of urticaria. Our findings may point to an IgE-independent urticaria pathway that could help address unmet clinical need.

[1] deCODE genetics/Amgen Inc., Reykjavik, Iceland. [2] Faculty of Medicine, School of Health Sciences, University of Iceland, Reykjavik, Iceland. [3] School of Science and Engineering, Reykjavik University, Reykjavik, Iceland. [4] School of Engineering and Natural Sciences, University of Iceland, Reykjavik, Iceland. [5] Department of Immunology, Landspitali, the National University Hospital of Iceland, Reykjavik, Iceland. [6] Department of Laboratory Hematology, Landspitali, the National University Hospital of Iceland, Reykjavik, Iceland. [7] Rheumatology Unit, Department of Medicine, Karolinska Institutet, Solna, Stockholm, Sweden. [8] Department of Medicine, Landspitali, the National University Hospital of Iceland, Reykjavik, Iceland. [9] The Medical Center Mjodd, Reykjavik, Iceland. [10] These authors contributed equally: Ragnar P. Kristjansson, Gudjon R. Oskarsson. ✉email: patrick.sulem@decode.is; kstefans@decode.is

Urticaria, also known as hives, is a skin disorder characterized by outbreaks of raised pruritic wheals caused by the release of histamine, typically ranging from a few millimeters to a few centimeters in diameter[1]. These outbreaks can occur as a reaction to allergens and physical stimuli or in an autoimmune or hereditary disorder, but the majority of outbreaks are idiopathic[2]. While most outbreaks resolve within 24 h, acute urticaria is defined as wheals lasting or recurring for less than six weeks, and longer-lasting recurrent outbreaks are classified as chronic[1,3,4]. Urticaria in any form is estimated to affect 15-25% of individuals, with chronic urticaria representing a rare form (0.5% in Europeans and 1.4% in Asians)[5], and can impact quality of life[6].

Mast cells and basophils in the skin play a key role in the pathophysiology of urticaria through the release of histamine and other mediators, including tryptases, into the skin, followed by vascular dilation and infiltration of lymphocytes and granulocytes into the lesions[7,8]. Sera from human urticaria patients as well as anti-IgE and anti-FcεRI antibodies have been shown to induce both basophil and mast cell degranulation[8,9]. Mast cells have been identified as the main effector cells in urticaria[10]. Low basophil count is also common in urticaria[11]. There is evidence that two mechanisms contribute to the pathogenesis of chronic urticaria; the development of autoantibodies to FcεRI or IgE that lead to degranulation of mast cells and basophils, reported to be responsible for 30–50% of cases, and dysregulation of intracellular signaling pathways[2]. Recent data indicate that both IgG- and IgE-specific autoantibodies, as well as autoantibodies against other surface antigens of mast cells and basophils may contribute to chronic urticaria[12].

Treatment of urticaria involves avoidance of external stimuli, and/or drug treatment, where antihistamines are the most commonly prescribed medication[2]. High dose antihistamines are only effective in about half of patients with chronic urticaria. The anti-IgE monoclonal antibodies omalizumab and ligelezumab are effective in antihistamine resistant chronic and spontaneous urticaria, but only increase treatment response to 65–80%[6,12–14]. Achieving full urticaria remission has been identified as an important clinical goal[6].

To search for sequence variants affecting risk of urticaria, we performed a meta-analysis of genome wide association studies (GWAS) on individuals with ICD-10 diagnostic code L50 and all sub-codes using data from Iceland, the UK, Finland, and Japan. The GWAS signals were further interpreted in the context of large-scale transcriptomics and proteomics data from Iceland and the UK.

## Results

We performed a meta-analysis of GWAS on 40,694 cases diagnosed with urticaria (ICD-10 code L50 and all sub-codes) and 1,230,001 controls from Iceland, the UK, Finland, and Japan. The meta-analysis was performed on associations under the additive model using a total of up to 26.7 million sequence variants. To adjust for multiple testing, we used a weighted Bonferroni correction based on variant annotation[15]. In the meta-analysis we detected associations of common sequence variants at nine loci with urticaria, (Table 1, Supplementary Data 1-2, Supplementary Figs. 1–10). Two of the urticaria associated sequence variants are predicted to be splice-donor variants in GCSAML and NFKB1. Two of the nine associations we detect have been reported with urticaria in individuals of East Asian and European descent[16] (Supplementary Data 3). Seven out of the nine urticaria-associated variants were tested in all four populations, but two variants were not tested in Japan (Table 1, Supplementary Data 1). No significant heterogeneity of effect between populations was observed after adjusting for multiple testing (Supplementary Data 1).

In the meta-analysis of urticaria, the most significant association, which also has the greatest effect, is between rs56043070[A], a predicted splice-donor variant in GCSAML (formerly C1orf150), (MAF ~ 6–7% in all cohorts; NM_145278.4:c.89+1G>A, Supplementary Data 4) and greater urticaria risk ($OR_{combined} = 1.24$, $P\text{-value}_{combined} = 3.6 \times 10^{-44}$; Table 1, Supplementary Data 1). GCSAML encodes Germinal Center-Associated Signaling And Motility-Like Protein, a poorly described protein named after its sequence similarity (30.3%) with GCSAM[17], an important factor for B cell receptor signaling[18,19]. The splice-donor variant correlates with a single variant in the intron of GCSAML; rs74227709[A] ($r^2 = 1.00$, $D' = 1$). At the GCSAML locus, we demonstrate co-localization of disease, biological trait, splice-QTL (sQTL), and protein-QTL (pQTL) signals from datasets generated in Iceland and the UK (Fig. 1a–g, Supplementary Figs. 2, 11–20, Table 2, Supplementary Data 5). The association of rs56043070[A] with urticaria was recently reported in a GWAS of 220 traits in Japan and was replicated in a cross-population meta-analysis from Finland and the UK[16].

The availability of individual genotypes in Iceland and the UK, allowed us to test the urticaria-associated variants under alternative models in the combined dataset from Iceland and the UK. Only the association under the full genotypic model with rs56043070[A] at GCSAML deviated significantly from the additive model ($P\text{-value}_{full\ vs.\ additive} = 1.5 \times 10^{-5}$, Supplementary Data 2). This indicates an increased risk of urticaria among both heterozygotes and homozygotes, and the risk conferred on homozygotes (genotypic frequency = 0.45%) is greater than expected under the additive model.

Then we tested the GCSAML variant for associations with all tested phenotypes under the full genotypic model (Table 2, Supplementary Data 5). The associations with platelet count and basophil percentage showed significant deviation from the additive model as well.

**Phenome analysis.** We tested the nine urticaria-associated variants with four inflammatory phenotypes (atopic dermatitis, asthma, allergic rhinitis, and psoriasis) in meta-analyses of cases and controls from Iceland, the UK, Finland, and Japan (Supplementary Data 6). We also tested for association between the 18 quantitative blood traits (cell counts, red cell characteristics, platelet characteristics) in a meta-analysis from Iceland, the UK, and Japan when available, and serum IgE levels in Iceland (total of 19 traits, Supplementary Data 7). We also tested the marker at GCSAML under the full model for the same phenotypes, given the deviation from additive model that we observe for urticaria. This resulted in 231 tests and we detected 53 associations given the number of tests ($P\text{-value} < 0.05/231 = 2.2 \times 10^{-4}$).

Consistent with previous reports, we observed associations of variant rs3024971[G] at STAT6 with three out of the four tested inflammatory phenotypes (asthma, allergic rhinitis, and atopic dermatitis), where the minor allele associates with protection against these phenotypes in addition to urticaria (Supplementary Data 6, 8). Of note, no associations were observed for the GCSAML variant with the four tested inflammatory phenotypes despite a rather large effect on urticaria.

For variants at the FCER1A and the STAT6 loci, the allele that associates with lower risk of urticaria also associates with decreased serum IgE levels and both are highly correlated with variants previously reported to associate with serum IgE levels ($r^2$ of 0.87 and 1.00, respectively) (Supplementary Data 7, 8)[20,21]. The allele of the FCER1A variant that associates with a lower risk of urticaria also associates with decreased risk of asthma, given the

**Table 1 Associations of sequence variants with urticaria in a meta-analysis of GWAS from Iceland, the UK, Finland, and Japan under the additive model.**

| Marker (position) | Allele[a] (min/maj) | MAF_Ice,UK,Fin,Jap (%) | Gene/[Locus] | Evidence | Variant annotation | LD Class (r²>0.8) | OR_Ice,UK,Fin,Jap | P-value_Ice,UK,Fin,Jap | Combined OR (95% CI) | P value | Phet |
|---|---|---|---|---|---|---|---|---|---|---|---|
| rs56043070 (chr1:247556467) | A/G | 6.6, 7.0, 5.4, 6.2 | GCSAML | sQTL | Splice donor | 2 | 1.25, 1.23, 1.24, 1.24 | 3.1E-13, 3.2E-13, 1.3E-10, 6.9E-12 | 1.24 (1.20-1.28) | 3.6E-44 | 0.97 |
| rs6703348 (chr1:159321893) | G/C | 28.8, 25.7, 26.9, 6.3 | FCER1A | eQTL | Intergenic | 9 | 0.91, 0.95, 0.90, 0.90 | 2.1E-7, 0.0048, 7.3E-9, 5.8E-04 | 0.92 (0.90-0.94) | 8.2E-18 | 0.10 |
| rs386480 (chr6:31979060) | G/C | 34.9, 31.0, 28.2, 30.4 | C4A | eQTL, sQTL | Upstream | 9 | 0.92, 0.93, 0.91, 0.98 | 1.3E-6, 4.6E-5, 8.1E-8, 0.13 | 0.93 (0.92-0.95) | 6.2E-15 | 0.02 |
| rs143871515 (chr3:105704804) | CTAAT/C | 32.6, 27.5, 35.6, 22.7 | CBLB | eQTL | Intron | 90 | 0.93, 0.93, 0.92, NA | 1.3E-5, 2.1E-5, 3.2E-6, NA | 0.93 (0.91-0.94) | 1.8E-14 | 0.97 |
| rs56404800 (chr16:88449304) | A/T | 28.3, 31.7, 27.9, 48.8 | ZFPM1 | | Upstream | 33 | 0.93, 0.96, 0.92, 0.95 | 6.1E-5, 0.012, 9.4E-6, 2.7E-4 | 0.94 (0.92-0.96) | 4.8E-13 | 0.38 |
| rs12493005 (chr3:177195111) | T/C | 41.0, 43.7, 37.8, 74.5 | TBL1XR1 | sQTL | Intron | 164 | 1.05, 1.04, 1.09, 1.06 | 0.0010, 0.0080, 1.5E-6, 9.9E-4 | 1.06 (1.04-1.08) | 2.9E-12 | 0.38 |
| rs4410077 (chr16:1253082) | T/C | 46.1, 41.6, 54.2, 79.2 | TPSD1 | eQTL, pQTL | Upstream | 8 | 1.05, 1.07, 1.08, NA | 0.0042, 4.6E-5, 8.2E-7, NA | 1.07 (1.05-1.09) | 7.0E-12 | 0.35 |
| rs3024971 (chr12:57099944) | G/T | 10.3, 10.4, 4.4, 2.9 | STAT6 | sQTL | Downstream | 2 | 0.92, 0.91, 0.89, 0.90 | 0.0022, 1.3E-4, 0.0048, 0.019 | 0.91 (0.88-0.94) | 1.2E-09 | 0.93 |
| rs2272676 (chr4:102502169) | T/G | 34.4, 34.1, 34.9, 32.2 | NFKB1 | eQTL, sQTL, pQTL | Splice donor | 75 | 1.05, 1.05, 1.05, 1.04 | 0.0092, 0.0043, 0.0049, 0.012 | 1.05 (1.03-1.06) | 7.3E-08 | 0.99 |

The table shows MAF, OR, and P-value for each population. Iceland_Ncase/Ncontrol: 14,312/354,647; UK_Ncase/Ncontrol: 8,730/399,923; Finland_Ncase/Ncontrol: 7,759/313,241; Japan_Ncase/Ncontrol: 9,893/162,190. MAF, OR, and P-value are shown for each population. Effects are presented for the minor allele.
MAF minor allele frequency, LD linkage disequilibrium, LD-class size total number of variants correlating with the variant (r² > 0.8), CI confidence interval, Phet P value for test of heterogeneity between the ORs in tested population, Coding variant correlates with a coding variant (r² > 0.8), eQTL variant correlates with top cis expression quantitative trait loci (eQTL) (r² > 0.8), sQTL variant correlates with top splice QTL (r² > 0.8), pQTL variant correlates with top cis protein QTL (r² > 0.8).
[a]Minor/major allele in Europe.

number of tests (OR = 0.974, $P = 2.9 \times 10^{-6}$) (Supplementary Data 6).

Notably, the *GCSAML* splice-donor rs56043070[A] variant does not associate with IgE levels. However, it is associated with basophil percentage of white blood cells (WBC) driven by homozygotes (Effect_het = 0.001 SD, Effect_hom = −0.22 SD, full model *P*-value = 1.0. × $10^{-27}$; n = 665,329; Table 2, Supplementary Data 5, 7, Supplementary Fig. 21). The urticaria risk-allele associates with a reduction in basophil percentage of white blood cells, consistent with low basophil count in chronic urticaria[22].

We explored the genetic correlation between urticaria (ICD-10 code L50) diagnosis and 1985 case-control phenotypes defined from ICD codes and 18 quantitative phenotypes in the UK dataset (Serum IgE levels were not available from the UK). After correcting for multiple testing (significance thresholds: $P < 0.05/2003 = 2.5 \times 10^{-5}$), we found a positive genetic correlation between urticaria (ICD-10 code L50) and asthma (ICD-10 code J45) ($r_g = 0.38$, $P = 1.2 \times 10^{-5}$) (Supplementary Data 9, 10), but not with any other diseases or traits.

**RNA analysis and plasma proteomics.** In order for us to gain insight into which genes and proteins are involved in the pathogenesis of urticaria, we looked for a strong correlation ($r^2 > 0.80$) between the urticaria-associated variants and coding or splice variants, top cis- pQTLs (based on 4792 plasma proteins measured in 35,559 Icelanders using SOMAscan and 3072 plasma proteins measured in 47,000 individuals from the UK using Olink), top cis-eQTLs (based on RNA sequencing analysis of blood from 17,848 Icelanders and from a set of eQTL databases summary statistics), and top sQTLs (based on same Icelandic RNA sequencing set).

Our data reveal that eight of the nine urticaria-associated variants correlate with at least one of the above. Two of the variants correlate with cis-pQTLs, five with cis-eQTLs, and five with cis-sQTLs (Supplementary Data 11–14).

*GCSAML.* The most significant urticaria-associated variant, the predicted splice-donor variant at the *GCSAML* locus, is located at the first base pair of intron 2 in the primary transcript of *GCSAML* (ENST00000366488.5, Supplementary Data 4, Fig. 2), disrupting the canonical splice-donor motif (GT becomes AT). This disruption does not affect the total expression of *GCSAML* (Effect = −0.08 SD, *P*-value = 0.45, Supplementary Data 12, 13), but leads to a shift in transcript usage (Supplementary Figs. 20, 22, Fig. 1D). Heterozygotes have 43% lower expression of the primary transcript than non-carriers, while the two homozygous carriers of the variant do not express the primary transcript at all (Fig. 2). We observe a concomitant increase in expression of a novel transcript that splices out exon 2 of the primary transcript (*P*-value = 2.7 × $10^{-238}$; Fig. 2, Supplementary Figs. 23–27; Methods). This novel transcript preserves the reading frame, as exon 2 is 60 base pairs long. Since the coding sequence of exon 1 is 29 base pairs long, the variant also leads to a serine to arginine amino acid substitution at the exon-exon boundary (Fig. 2). Carriers of rs56043070[A] also expressed more transcripts retaining intron 2 (*P*-value = 1.6 × $10^{-20}$; Supplementary Figs. 24–27).

We found significant *trans*-pQTL associations of the *GCSAML* urticaria-associated variant with plasma levels of five proteins encoded by *TPSAB1*, *TPSB2*, *KIT*, *SELP*, and *SIGLEC6* (*P*-values < 1.0 × $10^{-7}$; Supplementary Data 11). *TPSAB1* encodes tryptase alpha/beta-1, *TPSB2* encodes tryptase beta-2, *KIT* encodes mast/stem cell factor receptor, *SELP* encodes P-selectin, and *SIGLEC6* encodes sialic acid binding Ig-like lectin 6. All five proteins are linked to mast cell activation, which is at the

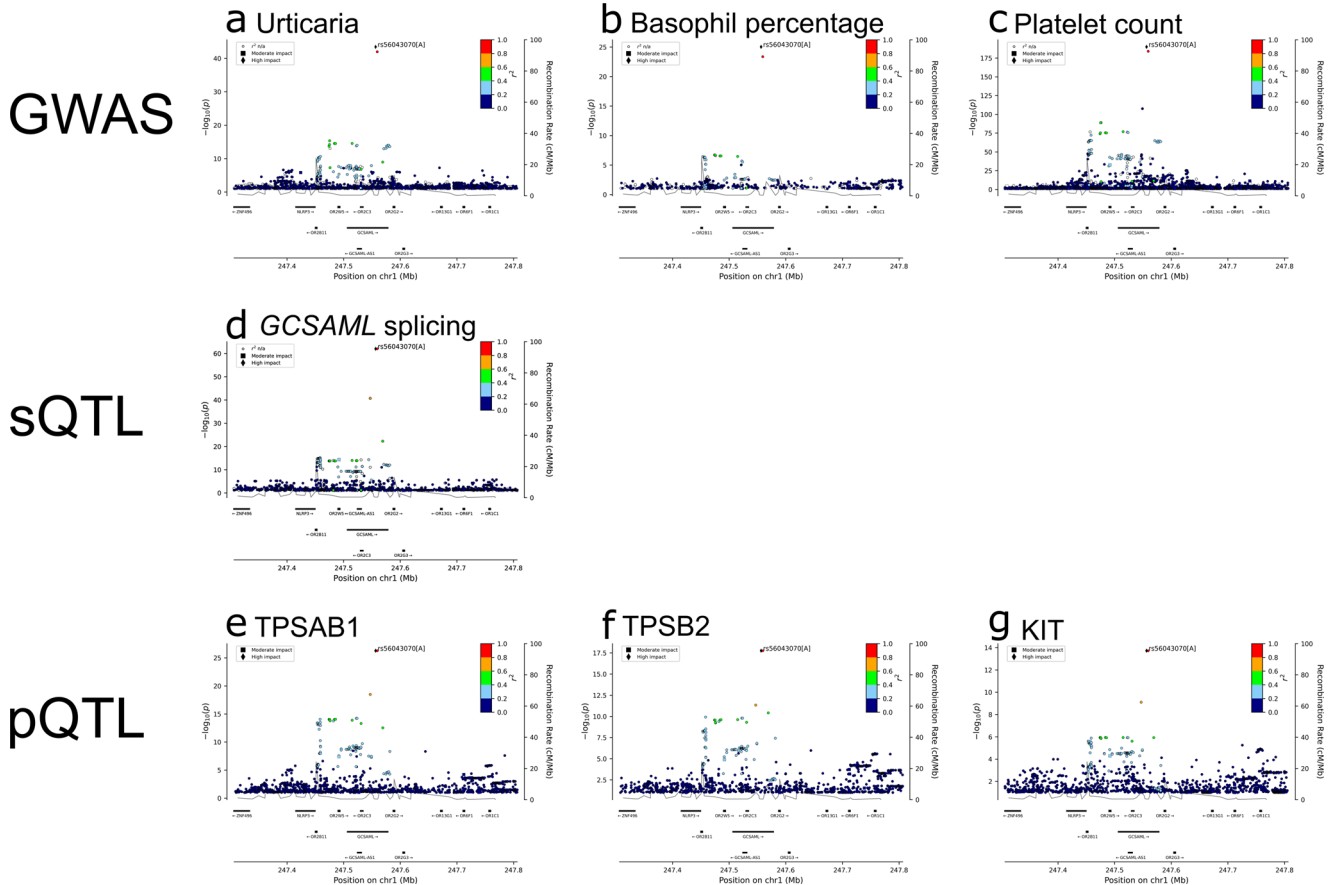

**Fig. 1 Co-localization of signals at the GCSAML locus.** The splice donor variant rs56043070[A] is the strongest-associating variant at this locus for the reported GWAS, sQTL, and pQTL signals. The splice-donor variant is indicated by its rs number, and other variants are colored according to correlation ($r^2$) with it (legend at top-right of each panel). Correlation between variants is estimated using genotype data from the Icelandic population. $-\log_{10}P$ values of depicted variants are shown along the left y-axis of each panel. The right y-axis of each panel shows calculated recombination rates at the chromosomal location, plotted as a solid black line. For all individual locus plots, see Supplementary Figs. 2, 11–17. **a** The sequence variant at *GCSAML* associated with increased risk of urticaria under the additive model in the meta-analysis of GWAS from Iceland, the UK, Finland, and Japan. **b** The variant at *GCSAML* associating with decreased basophil percentage of white blood cells under the recessive model in the meta-analysis of GWAS from Iceland and the UK. **c** The variant at *GCSAML* associated with decreased platelet count under the additive model in the meta-analysis of GWAS from Iceland, the UK, Finland, and Japan. **d** The sQTL signal at *GCSAML* associating with RNA splicing of *GCSAML* in adipose tissue in Iceland. **e** The pQTL signal at *GCSAML* associating with plasma levels of *TPSAB1* protein products in Iceland. **f** The pQTL signal at *GCSAML* associating with plasma levels of *TPSB2* protein products in Iceland. **g** The pQTL signal at *GCSAML* associating with plasma levels of *KIT* protein products in Iceland.

**Table 2 Association of the splice-donor variant rs56043070[A] in *GCSAML* with urticaria and related traits under the full genotypic model in the combined Icelandic and UK dataset.**

| Phenotype | OR/Beta_hetero (95% CI) | OR/Beta_homo (95% CI) | P-value | P-value full model vs additive |
|---|---|---|---|---|
| Urticaria | 1.17 (1.12, 1.23) | 2.17 (1.80, 2.63) | $2.9 \times 10^{-27}$ | $1.5 \times 10^{-5}$ |
| Basophil percentage | 0.00[a] (-0.01, 0.01) | −0.22[a] (−0.26, −0.18) | $1.0 \times 10^{-27}$ | $2.7 \times 10^{-23}$ |
| Platelet count | −0.11[a] (−0.12, −0.11) | −0.42[a] (−0.46, −0.39) | $2.2 \times 10^{-322}$ | $8.2 \times 10^{-191}$ |
| Plasma TPSAB1 levels[b] | 0.16[a] (0.13, 0.19) | 0.55[a] (0.39, 0.71) | $6.2 \times 10^{-33}$ | 0.0061 |
| Plasma TPSB2 levels[b] | 0.11[a] (0.08, 0.14) | 0.40[a] (0.26, 0.54) | $4.8 \times 10^{-23}$ | 0.038 |
| Plasma KIT levels[b] | 0.14[a] (0.11, 0.17) | 0.10[a] (−0.06, 0.26) | $5.8 \times 10^{-15}$ | 0.079 |

Effects are presented for the genotype based on the minor allele, compared to non-carriers.
*OR/Beta_hetero* odds ratio/Beta for heterozygous carriers, *OR/Beta_homo* odds ratio/Beta for homozygous carriers, *CI* confidence interval, *P-value full vs. additive* the P-value for test of heterogeneity between the ORs/betas under the full genotypic model and the additive model. For association results for the individual populations, see Supplementary Data 5.
[a]Betas are presented in standard deviations (SD).
[b]Only tested in the Icelandic data.

center of the pathogenesis of urticaria[23–25]. In all five cases, the urticaria risk allele associated with higher protein levels (Supplementary Data 11). Plasma levels of the *TPSAB1* and *TPSB2* protein products increase with rs56043070[A] allele count regardless of disease status (Supplementary Fig. 28), suggesting

that the observed increased plasma protein levels are driven by genotype independently of disease status.

*FCER1A*. In the Icelandic data, the urticaria-associated variant rs6703348[G] at *FCER1A* is highly correlated with two top

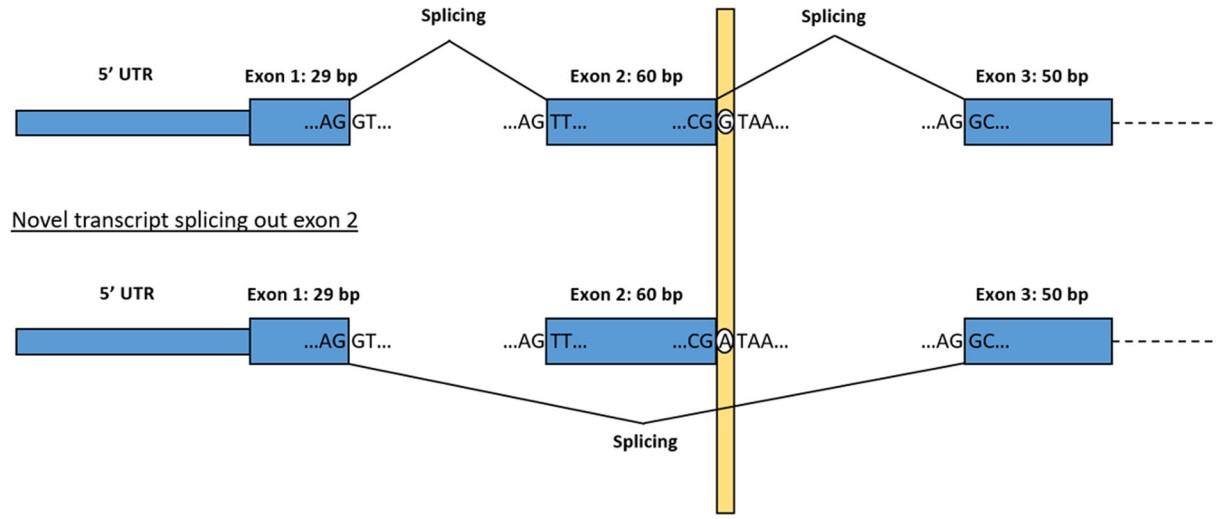

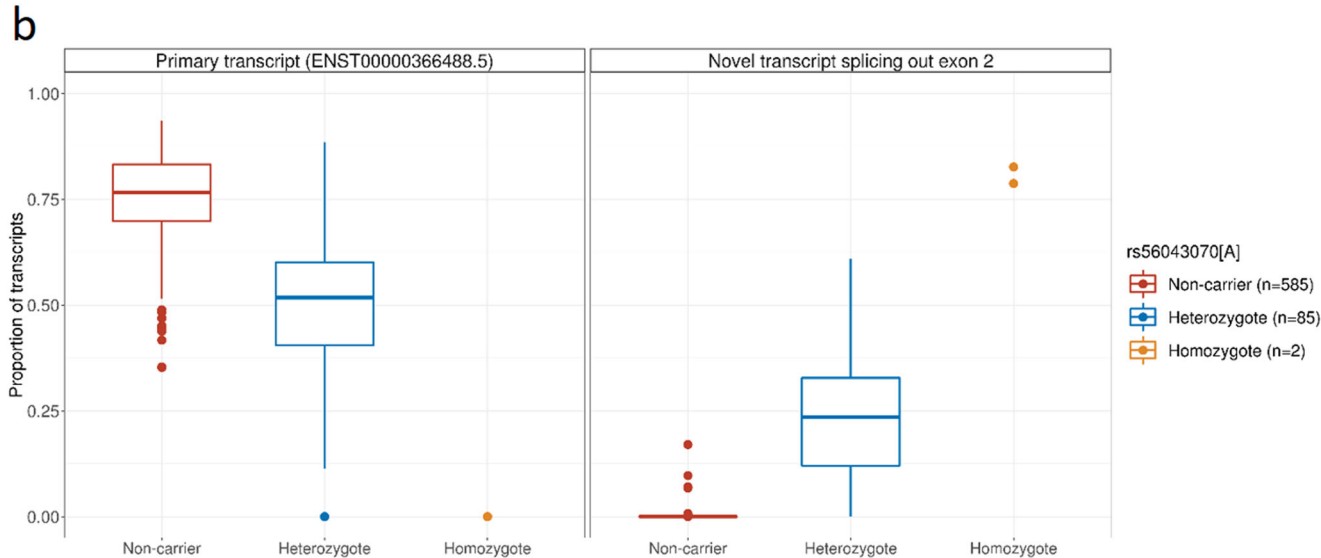

**Fig. 2 The effect of the splice-donor variant rs56043070[A] in *GCSAML* on RNA splicing in subcutaneous adipose tissue. a** A schematic illustration of alternative splicing of the first three exons in *GCSAML*, enumerated according to the primary transcript (ENST00000366488.5), among carriers and non-carriers of rs56043070[A]. The variant is located at the first base pair of intron 2, disrupting the conserved splice donor site motif (GT becomes AT). **b** Boxplot of the quantified proportions of two *GCSAML* transcripts; the primary transcript (ENST00000366488.5) and the novel transcript splicing out exon 2, stratified by rs56043070 genotype. After removing 78 samples due to lack of expression of the first three exons, we used 672 samples for the analysis. The hinges of the boxes represent the quartiles of the distribution, and whiskers represent the highest and lowest point no further than 1.5 times the interquartile range from the upper and lower hinges, respectively. For more detail, see Supplementary Figs. 10–15 and Supplementary Data 16.

eQTLs for RNA levels of *FCER1A* in blood and in monocytes ($r^2 = 0.98$ and 0.82, respectively) (Supplementary Data 15-16). In both cases, the allele associated with increased *FCER1A* RNA levels in blood is also associated with decreased risk of urticaria, like rs6703348[G]. The variant is also associated with decreased risk of asthma and lower serum IgE levels (Supplementary Data 8-10). These results are consistent with reported associations of a correlated sequence variant at the locus (rs2427837[G], $r^2 = 0.84$) with asthma, *FCER1A* RNA levels, and IgE levels[26]. The variant rs6703348[G] is also correlated with top *cis*-eQTLs for RNA levels of *FCER1A* in other tissues (Supplementary Data 12-13). *FCER1A* encodes the alpha subunit of the high-affinity IgE receptor (FcεRI), a protein that binds to the Fc region of IgE and mediates the activation of mast cells and basophils[2,27].

*Other loci.* Intron variant rs3024971[G], at the *STAT6* locus, is the top *trans*-pQTL in the Icelandic data for plasma levels of inter-leukin 5 receptor alpha subunit, encoded by *IL5RA* (Supplementary Data 11). IL5RA plays an important role in coordinating the release of eosinophil and IgE production and it has been suggested to be involved in the pathogenesis of asthma[28]. This is consistent with our and previous observation of the association between this *STAT6* variant and asthma. *STAT6* encodes signal transducer and activator of transcription 6, part of a family of proteins responsible for transmitting signals into the nucleus and thus regulating gene transcription[29]. The *STAT6* intron variant is highly correlated with the top-sQTL ($r^2 = 1.0$), where it is implicated in alternative splicing of *STAT6* on RNA-sequencing (Supplementary Data 14). Carriers of the minor allele of the sQTL partially retains intron 14 and 15 (Supplementary Fig. 29).

The urticaria risk allele of the upstream gene variant rs4410077[T], at *TPSD1*, positively correlates with the variant rs8051930[A] ($r^2 = 0.84$) associating with increased plasma levels of TPSAB1 (pQTL), and higher RNA expression levels of *TPSD1* (eQTL), and *TPSB2* (eQTL) in blood, all at the same locus coding tryptases (Supplementary Data 11–14). The variant associates with no other phenotypes. Tryptases are secreted by mast cells and are involved in the pathogenesis of allergic diseases, including urticaria. *TPSD1* is located at the tryptase locus and an increased *TPSAB1* copy number has been associated with elevated basal serum tryptase levels and a multisystem disorder[30]. The urticaria-associated variant rs4410077[T] at *TPSD1*, does not correlate with a reported copy number of *TPSAB1* ($r^2 = 0.011$).

At *NFKB1*, the risk allele for urticaria positively correlates with variants associated with higher RNA expression in neutrophils and adipose ($r^2 > 0.95$) and increased plasma NFKB1 protein levels (rs230539[G]; $r^2 = 0.96$) (Supplementary Data 11–14). Sequence variant rs2272676[T] at *NFKB1* is also highly correlated ($r^2 = 1.0$) with top sQTL, depicted in increased proportional usage of *NFKB1* transcript ENST00000505458 (Supplementary Data 14, Supplementary Fig. 30). *NFKB1* encodes a Rel protein-specific transcription factor (p105), which is also a precursor for a subunit (p50) of the ubiquitous NF-κB transcription factor. NF-κB is known to be involved in inflammatory pathways and in pathogenesis of auto-immune disorders[25,31,32].

## Discussion

Using a three-layered approach, including genomics, transcriptomics, and proteomics, we have discovered associations of nine sequence variants associating with urticaria in a meta-analysis of GWAS from Iceland, the UK, Finland, and Japan, and characterized their effect on other phenotypes.

Six of our urticaria-associated variants are pointing to genes that are directly implicated in type 2 immune response and/or mast cell physiology (*CBLB*, *FCER1A*, *GCSAML*, *STAT6*, *TPSD1*, *ZFPM1*)[33–35], one in the complement innate immunity (*C4A*), and one involved in NF-κB signaling, a transcription factor known to contribute to the pathogenesis of various inflammatory diseases when deregulated[36].

The most notable sequence variant we found to associate with urticaria is a splice-donor variant in *GCSAML*. We clearly implicate *GCSAML* in the pathophysiology of urticaria and in basophil numbers.

The ICD-10 code used in this GWAS study, L50, includes various sub-types of urticaria depending on a defined trigger. These include allergic, physical, and idiopathic urticaria, both chronic and acute. Even if a trigger can sometimes be identified, a large majority of urticaria is idiopathic. Detailed sub-phenotypes and duration of symptoms were not available in this study and it would be of interest to assess the effects of the identified variants on a well-characterized urticaria set to see if the effects differ on the different sub-types of urticaria.

Mast cells have been described as the primary effector cells in urticaria[10]. *GCSAML* expression is reported to be enriched in basophils and the HMC-1 mast cell line[37,38]. Furthermore, *GCSAML* was identified as one of eight novel mast cell-specific genes in an analysis of the human mast cell transcriptome ($n = 6$ individuals) in FANTOM5[39]. Expression levels were shown to be up to 20 times greater in mast cells than in basophils, the cell type with the second highest expression[39]. The association of the splice-donor variant in *GCSAML* with urticaria is thus congruent with the high *GCSAML* RNA expression seen in human mast cells.

We have confirmed the predicted effect of rs56043070[A] on *GCSAML* splicing, and demonstrated an alternative transcript usage. The variant leads to a significant increase in expression of a novel *GCSAML* transcript lacking exon 2, an exon that overlaps the first 4 amino acids of a 25 amino acid coiled-coil motif in GCSAML[40]. This increase is accompanied by a concomitant decrease in the levels of the primary transcript. It is unresolved, however, whether the associations we observe are caused by the reduced number of the primary transcript or by generation of the novel transcript. We note that the effects of rs56043070[A] on RNA splicing in adipose tissue gives limited information. The relevance in urticaria pathology is therefore not obvious. However, our data show that the splice-donor variant is capable of altering the splicing of *GCSAML* transcripts. It would be of interest to study this further in more relevant tissues or cell lineages, such as skin tissue and mast cells[10] to assess the relevance to urticaria pathology.

Taking advantage of proteomics data, we show that the *GCSAML* urticaria risk-allele associates with greater plasma levels of five proteins; TPSAB1, TPSB2, KIT, SIGLEC6, and SELP. TPSAB1 and TPSB2 both encode tryptases, while KIT encodes mast/stem cell factor receptor (SCFR). Tryptases are produced and released by both mast cells and basophils, but as the overall contribution of basophils is small, tryptases are considered markers of mast cell activation[41]. Similarly, the ligand of SCFR has been shown to play a key role in mast cell differentiation and activation[42,43]. *SIGLEC6* encodes an inhibitory motif (ITIM)-bearing receptor inhibiting mast cell activation through cross-linking with FcεRIα[44]. *SELP* encodes P-selectin, a cell adhesion molecule expressed on endothelial cells to attract leukocytes to site of inflammation, and is upregulated during mast cell activation[45]. TPSAB1, TPSB2, and KIT transcripts were shown to be enriched in mast cells (two-fold or greater expression than in any other cell type)[46]. Furthermore, the transcriptome-wide analysis of the FANTOM5 dataset identified *GCSAML*, *FCER1A*, *TPSAB1*, *TPSB2*, *KIT*, and *SIGLEC6* as either exclusively or preferentially expressed in mast cells[39]. The association profile of this variant, including all protein products significantly affected by it, indicates that *GCSAML* plays a role in mast cell regulation, with the splice-donor variant leading to greater mast cell activation.

Testing the same set of variants across all datasets allows us to demonstrate co-localization of signals at the *GCSAML* locus. The splice-donor variant is consistently the variant at this locus that most significantly associates with urticaria, basophil percentage, platelet count, RNA splicing of *GCSAML*, and levels of the five mast cell proteins. Together, these results demonstrate that *GCSAML* itself is important in the pathogenesis of urticaria.

Furthermore, our data are consistent with the causative nature of IgE in urticaria, i.e., the variants associating with decreased IgE levels also associated with decreased risk of urticaria. However, seven of the urticaria-associated variants did not associate with IgE levels, suggesting that its effect on urticaria may be mediated through another mechanisms. As current anti-histamine and anti-IgE monoclonal antibody treatments are ineffective in up to 35% of cases[12,13], further study of these pathways may help in addressing this unmet pharmacological need.

In this study, we combine genomics, transcriptomics, and proteomics in an unprecedented manner. Taken together, our results underpin the role of mast cells and type 2 inflammation in the pathogenesis of urticaria. We demonstrate how these powerful -omics tools can aid in linking variants to disease mechanisms, and more definitively profile the consequences of human sequence variation.

## Methods

**Ethics statement**. All Icelandic data were collected through studies approved by the National Bioethics Committee (approval no. VSN_19-157 or earlier VSN_14-099) following review by the Icelandic Data Protection Authority. Participants

donated blood or buccal samples after signing a broad informed consent allowing the use of their samples and data in all projects at deCODE genetics approved by the NBC. All personal identifiers of the participants' data were encrypted by a third-party system, approved, and monitored by the Icelandic Data Protection Authority.

The UK Biobank data were obtained under application number 56270. All phenotype and genotype data were collected following an informed consent obtained from all participants. The North West Research Ethics Committee reviewed and approved UK Biobank's scientific protocol and operational procedures (REC Reference Number: 06/MRE08/65).

Samples and phenotype data were collected from the Finnish biobanks and the national health registers, respectively, for the FinnGen database. The Coordinating Ethics Committee of the Hospital District of Helsinki and Uusimaa evaluated and approved the FinnGen study protocol. The project complies with existing legislation (in particular the Biobank Law and the Personal Data Act). The official data controller of the study is University of Helsinki.

Biobank Japan collected phenotype data (medical history and clinical data) and serum samples at participating medical institutions in Japan. All participants provided written, informed consent and the study was approved by ethics committees of the Institute of Medical Sciences, the University of Tokyo and RIKEN Center for Integrative Medical Sciences.

**Study subjects**. To identify urticaria cases in Iceland, we searched for patients with International Classification of Diseases (ICD-10) diagnosis code L50 and all sub-codes at Icelandic primary and secondary care centers. L50 covers various types of urticaria, i.e. allergic, physical, and idiopathic urticaria. Records spanned from 1985-2018.

The UK Biobank study is a large prospective cohort study of ~500,000 individuals in the age range of 40–69 from across the UK. UK urticaria cases were identified by extracting cases with ICD-10 code L50 and all subcodes from UK Biobank GP clinical event records (Field ID 42040) and hospital diagnoses made during inpatient admissions (Field IDs 41270 and 41271). Only British individuals of European ancestry were included in the study. Individuals in the urticaria cohort were excluded from the control group.

The phenotype data from the FinnGen study was produced from several national health registries. All urticaria cases were diagnosed by a physician and categorized using ICD-10 code L50 and all subcodes, ICD-9 code 708 and all subcodes, and ICD-8 code 708 and all subcodes. The summary statistics for available phenotypes, including urticaria, were imported on June 19th 2022 from a source available to consortium partners (version 7; http://r7.finngen.fi).

The phenotype data from Biobank Japan was gathered at 12 medical institutions in Japan where all study participants had been diagnosed with one or more of 47 target diseases by physicians at the cooperating hospitals and classified using ICD-10 codes. For urticaria cases, individuals with ICD-10 codes L50 and all subcodes were extracted from the data. The summary statistics from GWAS from Biobank Japan, including urticaria, were imported on October 10, 2020 from the Biobank Japan's website (https://pheweb.jp/).

We combined 14,312 cases and 354,647 controls from Iceland, 8730 cases and 399,923 controls from the UK, 7759 cases and 313,241 controls from Finland, 9893 cases and 162,190 controls from Japan; in total, 40,694 cases and 1,230,001 controls.

**Genotyping**. The Icelandic part of the study is based on testing 32,542,746 variants identified in whole-genome sequence (WGS) data from 28,075 Icelanders participating in various disease projects at deCODE genetics, sequenced using Illumina standard TruSeq methodology to an average genome-wide coverage of 37X. SNPs and indels were identified and their genotypes called using joint calling with Graphtyper[47]. The effects of sequence variants on protein-coding genes were annotated using the Variant Effect Predictor (VEP) using protein-coding transcripts from RefSeq. We carried out chip typing of 155,250 Icelanders (around 50% of the population) using Illumina SNP arrays. The chip-typed individuals were long-range phased[48], and the variants identified in the whole-genome sequencing of Icelanders were imputed into the chip-typed individuals. In addition, genotype probabilities for 285,644 untyped close relatives of chip-typed individuals were calculated based on Icelandic genealogy[49,50].

Genotyping of UK biobank samples was performed using a custom-made Affymetrix chip, UK BiLEVE Axiom[51], and with the Affymetrix UK Biobank Axiom array[52]. Imputation was performed by the Wellcome Trust Centre for Human Genetics using variants identified from 32,488 WGS individuals in the Haplotype Reference Consortium (HRC), UK10K haplotype resources, and 1000Genomes phase 3 panels[52,53]. This yielded a total of 96 million imputed variants, although only 40 million variants were imputed into 408,658 participants using the HRC reference set due to quality issues with the remaining variants.

FinnGen samples of 321,464 individuals were genotyped using Illumina and Affymetrix arrays (Illumina Inc., San Diego, and Thermo Fisher Scientific, Santa Clara, Ca, USA) (>650,000 SNPs). Genotype imputation of ~17 million variants was performed using a population-specific imputation reference panel comprised of 3775 whole genomes.

Biobank Japan genotyped 178,726 participants of East Asian ancestry using a combination of Illumina arrays. The genotypes were imputed using ~13.5 million

variants from WGS individuals in the 1000 Genomes Project Phase 3 version 5 genotype data ($n = 2504$) and Japanese whole-genome sequencing data ($n = 1037$).

**Statistics and reproducibility**. Logistic regression was used to test for association between variants and disease in Iceland and the UK using software developed at deCODE genetics[49]. We used two different models for association testing; the additive and the full genotypic model. In Iceland, sex, county of birth, current age or age at death (first and second order terms included), blood sample availability for the individual and an indicator function for the overlap of the lifetime of the individual with the time span of phenotype collection were included as covariates in the logistic regression model. We used county of birth as a proxy covariate for the first principal components in our analysis because county of birth has been shown to be in concordance with the first principal components in Iceland[54]. For the UK, sex, age, and the first 40 principal components were included as covariates. In Iceland and the UK, quantitative traits were tested using a linear mixed model implemented in BOLT-LMM[55]. The quantitative traits were adjusted for age and sex in both populations and also for 40 principal components in the UK Biobank. The traits were transformed to a standard normal distribution. Only British individuals of European ancestry were included from the UK Biobank. To account for cryptic relatedness and population stratification in the Icelandic and UK data, we used linkage disequilibrium (LD) score regression[56].

We adjusted the $\chi^2$ statistic from our GWAS scan by dividing it with the intercept (correction factor) from the LD score regression analysis. The correction factors that we obtained are consistent with the ones reported by others[57] (Supplementary Data 15). For urticaria, they were 1.02 for the UK and 1.10 for Iceland. For urticaria, we observed a high attenuation ratio of 0.71 for Iceland and 0.53 for the UK. The correction factor from LD score regression may be too high (attenuation bias) which may lead to over adjustment and conservative $P$-values[57]. Variants with imputation information below 0.8 were excluded from the analysis. We then combined the results from Iceland and the UK with imported association results from Finland and Japan to test for association between sequence variants and urticaria diagnosis (L50). The Finngen association analysis for binary traits was adjusted for sex, age, the genotyping batch, and the first 10 principal components. In the association analysis from Biobank Japan, both binary and quantitative traits were adjusted for age, sex, and the top 20 principal components as covariates.

For the meta-analysis, we used a fixed-effects inverse variance method based on effect estimates and standard errors from the four study groups for the additive model. For the full model from Iceland and the UK, we meta-analyzed using sample size and P-values[58]. Sequence variants were mapped to NCBI Build38 and matched on position and alleles to harmonize all four datasets.

We applied approximate conditional analysis, implemented in the GCTA software[59] to the meta-analysis summary statistics to identify independent association signals, using a stepwise model selection procedure. LD between variants was estimated using a set of 5000 whole-genome sequenced Icelandic individuals.

**Significance thresholds**. We applied genome-wide significance thresholds corrected for multiple testing using adjusted Bonferroni procedure weighted for variant classes and predicted functional impact. With 26,738,679 sequence variants being tested in the meta-analysis, the weights given in Sveinbjornsson et al., were rescaled to control the family-wise error rate (FWER)[15]. The adjusted significance thresholds are $2.9 \times 10^{-7}$ for variants with high impact ($N = 6263$), $5.8 \times 10^{-8}$ for variants with moderate impact ($N = 149,787$), $5.2 \times 10^{-9}$ for low-impact variants ($N = 2,165,903$), $2.6 \times 10^{-9}$ for other variants in Dnase I hypersensitivity sites ($N = 3,911,063$) and $8.7 \times 10^{-10}$ for all other variants ($N = 20,505,663$).

**Genetic correlation**. We calculated genetic correlations between urticaria (ICD-10 code L50) diagnoses and other traits as follows: We used cross-trait LD score regression and summary statistics for urticaria (ICD-10 code L50) in the deCODE dataset and from the UKB dataset for other traits. In these analyses, we used results for about 1.2 million well imputed variants, and for LD information we used precomputed LD scores for European populations (downloaded from: https://data.broadinstitute.org/alkesgroup/LDSCORE/eur_w_ld_chr.tar.bz2).

**Transcriptomics**. RNA sequencing was performed on subcutaneous adipose tissue ($n = 750$) and whole-blood ($n = 17,848$) from Icelanders. RNA sequencing reads were aligned to personalized genomes using the STAR software package[60], for splice-junction quantification and qualitative assessment of alignments (Supplementary Fig. 24). Gene expression was computed based on personalized transcript abundances estimated using kallisto with Ensembl v87 transcriptome reference[61]. Association between variants and gene expression was estimated using a generalized linear regression, assuming additive genetic effect and quantile normalized gene expression estimates, adjusting for measurements of sequencing artefacts, demographic variables, and leave-one-chromosome-out principal components of the gene-expression matrix. In addition, top cis-eQTL association results were collected from multiple publication and data sources listed in Supplementary Data 13[62-69]. We identified the top cis-eQTLs from all the eQTL data sources in deCODE sequence variant database and calculated genotypic correlation with all

nearby variants to determine if any of them were highly correlated ($r^2 > 0.80$) with the urticaria-associated GWAS variants.

To assess quantitatively the effect of the splice-donor rs56043070[A] on transcript usage in *GCSAML*, the transcriptome reference was extended to include novel transcripts splicing out exon 2 (enumerated according to the primary transcript ENST00000366488) and transcripts retaining intron 2 with and without rs56043070[A] variant (Supplementary Fig. 24). We re-analyzed the adipose RNA-sequences using kallisto[70] with the extended annotation, to accurately quantify the abundance of the novel transcripts. To obtain usage of primary and novel transcripts we aggregated transcripts into mutually exclusive groups; transcripts including exon 2, transcripts splicing out exon 2, and transcripts retaining intron 2. Due to lack of expression of the first three exons, we removed 78 samples from the analysis (Figs. 2, S11, S13 and S14). To assess the effect of nearby variants on *GCSAML* splicing, in particular skipping of exon 2, we associated all variants within a 500 kb window centered at the splice-donor variant with percentage spliced-in values (PSI) of the skipping of exon 2 using LeafCutter[71] (Figs. 1d and S17).

**Proteomics.** For Iceland, we tested the association of sequence variants with protein levels in plasma, using SomaLogic® SOMAscan proteomics assay. The assay scanned 4907 proteins in 35,559 Icelanders with genetic information available at deCODE genetics. Plasma protein levels were standardized and adjusted for age, sex, and year of sample collection (2000-2019).

From the UK we used measurements of 2938 protein levels from 48,684 UK Biobank participants measured using the Olink Explore 3072 platform[72]. Samples were selected stratified by age, sex, and recruitment center. Day of week of collection, participant ethnicity, and deprivation index were confirmed as representative of cohort distributions. Sample measurement and quality control were performed at Olink's facilities producing Normalized Protein eXpression (NPX) values for each protein per participant. NPX is Olink's relative protein quantification unit on log2 scale[73]. The plasma level measurements were standardized and adjusted for age, sex, and sample age.

**Reporting summary.** Further information on research design is available in the Nature Portfolio Reporting Summary linked to this article.

## Data availability

GWAS summary statistics for the meta-analysis of GWAS on urticaria are available and can be downloaded at deCODE genetics home page at https://www.decode.com/summarydata/. Sequence variants tested for association have been deposited in the European Variation Archive, accession number PRJEB15197. All other data are included in this publication or its Supplementary Data and are also available from the corresponding author (or other sources, as applicable) on reasonable request.

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

## Acknowledgements

The authors thank the individuals who participated in this study and whose contributions made this work possible. We also thank our valued colleagues who contributed to the data collection and phenotypic characterization of clinical samples as well as to the genotyping and analysis of the whole-genome association data. This research has been conducted using the UK Biobank Resource under application numbers 24711 and 24898.

## Author contributions

R.P.K., G.R.O., A.S., A.O., I.J., P.S., and K.S. designed the study and interpreted the results. R.P.K., G.R.O., A.O., V.T., G.B., G.T., B.R.L., P.T.O., U.S.B., T.O., I.J., and P.S. carried out the subject ascertainment and recruitment. R.P.K., G.R.O., S.R., S.H.L., E.L.S., G.H.E, G.H.H., S.K., K.J., B.V.H., S.S, P.M., G.L.N., and P.S. performed the sequencing, genotyping, and expression analyses. R.P.K., G.R.O., A.S., A.O., G.S., B.O.J., E.F., D.B., R.F., G.A.A., H.K., M.H.S., L.S., E.V.I., D.F.G., and P.S. performed the statistical and bioinformatics analyses. R.P.K., G.R.O., A.S., T.O., D.F.G., U.T., I.J., P.S., and K.S. drafted the manuscript. All authors contributed to the final version of the paper.

## Competing interests

Authors affiliated with deCODE genetics/Amgen Inc., R.P.K., G.R.O., A.S., A.O., S.R., G.S., S.H.L., B.O.J., E.L.S., G.H.H., E.F., D.B., S.K., K.J., R.F., G.A.A., H.K., M.H.S., V.T., L.S., E.V.I., G.B., B.V.H., G.T., S.S., P.M., G.L.N., T.O., D.F.G., U.T., I.J., P.S., and K.S. declare competing interests as employees. The remaining authors declare no competing interests.
