## [Peer Review File · Communications Biology]

Reviewers' comments:

Reviewer #1 (Remarks to the Author):

Sequence variant affects GCSAML splicing, mast cell specific proteins, and risk of urticaria

This paper face an unmet need in URTICARIA, it is the identification of some genetic hallmark that could help in the pathogenesis knowledge and further management of urticaria. The population studied is large and even two cohorts of people coming from two different countries were studied showing reproducible data. It's an interesting approach. There are some questions that should be taken into account directly related with the clinical knowledge and management of the urticaria. The authors are far from this clinical practice and the phenotype expression of the urticaria and the different trigger factors. Urticaria is managed as a unique disease in this paper and it is not. The entire paper should take into account this fact. In general the overview of urticaria is weak. It would be interesting to know what the authors think this new knowledge will impact in the general management of urticaria.

1. Urticaria is estimated to affect 15-25% of individuals

Please specify to which type of urticaria belong this percentage as is not the same when we refer to Acute Urticaria, to Chronic Urticaria, to Chronic Spontaneous Urticaria and to Chronic Inducible Urticaria .

2. WGS, please specify the meaning of this acronym in the first paragraph of the section "Results "

3. Methods – Study Subjects and Genotyping

According with the authors, to identify the urticaria patients was used the (ICD10) code L50, this code include all types of urticaria independently is acute, chronic spontaneous urticaria or inducible. L50 include, Allergic, cholinergic, chronic, due to food, due to heat, due to cold, due to plants, Dermographic, no specific, facticia (which is the same than Dermographic), idiopathic, non allergic, periodic recurrent, contact, solar, thermic (heat and cold), vibratory.

ICD was available but do not represent the actual classification of urticaria, but it is now a days the unique available until the ICD 11 will be implemented.

As L50 include different types of urticaria, acute and chronic, independently of the endogenous or exogenous trigger factor this should be well explained from the introduction. In the introduction of the paper the authors just mention the main and not unique mechanism described in Chronic Urticaria, spontaneous and inducible, the autoimmune hypothesis. This is mechanism is absolutely independent from the mechanism involved e.g. in the pure and minority allergic urticaria. The L50 ICD code, which is the source of this study, includes all classic phenotypes of acute and chronic urticaria. This is an important fact to be considered and should not be mixed as it will provide a confuse pathogenic approach to the different types of urticaria.

4. Methods - Association analysis

In this analysis, I understand that it was not considered the different types of urticaria available.

5. Please specify the donors signed which type of informed consent. If its a general informed consent for genetic studies, or specifically for this study

6. Results – GCSAML

The authors do an interesting exercise, looking for the differences between a single or recurrent episodes showing similar effects. And is good showing no differences according this course behavior. This approach does not take care of the true real life diagnosis o management of urticaria and its phenotypes. Although it was not the objective of this study this limitations should be taken into

account.

7. GCSAML and blood cell traits.

The authors associate the risk-allele with a reduction in basophils, that was well described in chronic urticaria.

At this point also different considerations should be taken into account. Basopenia in urticaria is not a generalized finding in general practice and it depends about how basophils and which type. As it is mention it has been described mainly in chronic spontaneous urticaria. I this study this subtype analysis was not performed. A fact to be discussed.

8. The author ignore in the paper previous Transcriptome studies done in urticaria as are the e.g. following :

Allergy Rhinol (Providence). 2015 Jan;6(2):101-10. doi: 10.2500/ar.2015.6.0124.
Gene expression profiles in chronic idiopathic (spontaneous) urticaria.
Patel OP1, Giorno RC, Dibbern DA, Andrews KY, Durairaj S, Dreskin SC.

Allergy. 2017 Nov;72(11):1778-1790. doi: 10.1111/all.13183. Epub 2017 May 26.
Transcriptome analysis of severely active chronic spontaneous urticaria shows an overall immunological skin involvement.
Giménez-Arnau A1, Curto-Barredo L1, Nonell L2, Puigdecamet E2, Yelamos J3, Gimeno R3, Rübberg S4, Santamaria-Babi L5, Pujol RM1.

Allergy. 2019 Jan;74(1):141-151. doi: 10.1111/all.13547. Epub 2018 Oct 15.
Omalizumab normalizes the gene expression signature of lesional skin in patients with chronic spontaneous urticaria: A randomized, double-blind, placebo-controlled study.
Metz M1, Torene R2, Kaiser S3, Beste MT2, Staubach P4, Bauer A5, Brehler R6, Gericke J1, Letzkus M3, Hartmann N3, Erpenbeck VJ3, Maurer M1.

9. A final consideration about what is the impact of these findings in the Urticaria knowledge would be interesting to know.

Reviewer #2 (Remarks to the Author):

General comments:

Authors reported GWAS of urticaria in an Iceland population. They reported a novel signal in GCSAML region, which was well replicated in independent cohort (UKBB). They added functional investigations by analyzing transcriptome- and proteome-wide analyses in the large Icelandic dataset; they showed an sQTL effect against GCSAML can explain the strongest GWAS association; and also showed the same variant is associated with plasma levels of tryptase, which is an important enzyme in angioedema. They also discussed signal similarity with GWAS of IgE. Overall, I appreciated the novelty and the importance of this study, and I believe this study improved our understanding of the etiology of urticaria. However, I think authors can improve the downstream analyses in several ways.

Major comments:

1, The effort to conduct a replication analysis in UKBB is a good strategy. However, since UKBB whole genetic information is now available, authors should be able to do GWAS of urticaria in UKBB and conduct meta-analysis for all variants (not limited to a few variants). In Table 1, I can see the meta-analysis statistics for only 4 variants. But I am curious to know the results of all other variants. I expect, they can find more novel signals from meta-analysis.

2, One key analysis missing in this manuscript is LD score regression analysis, which now becomes a standard approach in GWAS downstream analysis. It can be used in three purposes;

(i) Quality control of GWAS results: using intercept as an indicator of bias, authors should be able to provide how much of the genetic signals is bias (PMID: 25642630). It is a very important metric to evaluate the quality of GWAS results.

(ii) Genetic correlations (PMID: 26414676). In Figure S17, they nicely showed the correlations of genetic effects between IgE and urticaria. This is very interesting and important to understand its etiology. However, they should extend this analysis to other diseases, not limited to IgE. Using genetic correlation based on LDSC, they should be able to assess genetic relationship of urticaria not only with IgE but also with a variety of diseases.

(iii) Partitioning heritability (PMID: 26414678). They can partition urticaria heritability using LDSC. If there are any publicly available epigenomic annotations of cell types which they believe important for urticaria (including mast cells), they can infer enrichment of heritability within such annotations. This analysis could provide genetic evidence that such cell types are important for urticaria.

3, Related to sections of "Reported variants and urticaria"

- it is not clear how they selected "396 variants reported to associate with related immunological traits (L81)" just by reading the main text. Are these 396 variants the all variants reported for these traits (e.g. in GWAS catalog)? This selection strategy is very critical since they set significant threshold at 0.05/396.

- This is an analysis based on their hypothesis that variants associated with clinically related diseases (e.g. asthma) should also related to urticaria. This might be true, but they did not provide any genetic evidence to support their hypothesis. As I discussed above, they can add genetic correlation analysis to confirm their hypothesis: for example, they can ask their urticaria GWAS is significantly correlated with GWAS of asthma.

- Also, they should not introduce signals detected in this analysis as significant associations in GWAS (as in abstract and L83). FCER1A and STAT6 should not treated in same way as GCSAML and HLA because the robustness of this analysis is much weaker than that of GWAS.

Minor comments:

- In abstract, authors utilized different significant thresholds. It is very confusing. First, they reported "We found two sequence variants associating with urticaria". Next, they reported "we found two variants associating with urticaria". Apparently, they utilized two different significant thresholds. They have to make abstract more readable on this point.

- Also in abstract, they utilized expressions like "variants in FCER1A" in several places. However, variants in gene does not make sense. They should say "variants in FCER1A region" or "variants in FCER1A loci"

- L148 "copy number" should be replaced with "allele count". It is confusing with copy number variants.

- L153 "The association of this variant with urticaria replicates nominally in the UK (OR = 1.06, P-value = 154 0.01; Table 1)." They have to emphasize heterogeneity.

- Association pattern of rs56043070 is different in different traits. Some are well fit with additive model, others are not. Can they provide any reasonable speculations why this phenomenon was observed?

- Urticaria is a clinical diagnosis with multiple pathophysiology as they wrote in the introduction section. However, they did not clearly provide how they included patients with urticaria in the main text (L79). In addition to method, they should provide such critical information in the main text.

Reviewer #3 (Remarks to the Author):

This is a large GWAS of hives (urticaria) with an Icelandic discovery set and a UKBB derived replication set (total case N ~22,000, >750k controls). The Icelandic data set, as with other deCode analyses, was derived partially from sequencing and then imputation of untyped relatives. 4 loci were identified. These were followed up using transcriptomic and proteomics. The manner in which the trait was defined appeared appropriate. The results are likely to be of interest to people working in immunity/allergic disorders.

Major concerns

No ancestry/population sub structure correction in the logistic regression GWAS performed in the Icelandic GWAS. The potential for inflation of test statistics (e.g. LDscore intercept) need to be described and discussed in the main text, not in the final paragraph of sup page 3, and in sup table 15. Sup table 15 shows the corrective factor (LD score intercept) for the main Urticaria GWAS is very high for the Icelandic set- 1.10 - and should be addressed e.g. with principal components, and/or approaches that better handle population structure for binary traits such as SAIGE PMID: 30104761, not just corrected for without further investigation. The UK Biobank dataset does not seem to have the same issue for Urticaria GWAS (~1.03). Does the hives association for the lead 4 SNPs remain if population structure is corrected for in the Icelandic GWAS? I suspect they will still be associated, but this should be confirmed. If the authors cannot use SAIGE etc for binary GWAS it is possible to approximate a log(OR) from the bolt-lmm regression output e.g. see section 10.2 of the bolt-lmm guide at <https://data.broadinstitute.org/alkesgroup/BOLT-LMM/>.

Bolt-LMM was used for quantitative traits which should hopefully be addressing any sub structure for the quantitative measures; for BOLT-LMM analyses it is more informative to report the attenuation ratio rather than LDSC intercept (since this can be inflated in mixed models; see PMID: 29892013). Ideally here the attenuation ratio should be < 0.08, and if not this should be discussed/addressed.

The authors have used two models to test for association – additive and a full genotypic model. The full details for these models are in the supplemental, and that is probably okay given their length, but this should be referenced to in the main text method section.

The approach for setting genome-wide significant threshold (page 9 line 261) in the discovery set is appropriate but should be described in the main methods not in the sup. Does this threshold apply just to the Icelandic GWAS data or UKBB as well? See following as well

It is not clear which association model's P-value is used to identify loci that reach genome-wide significance in the discovery set – additive, full etc.

What replication means in this article is very unclear; as a result I have a series of sub questions related to this. Overall the presentation of results for replication is hard to match up with the abstract/introduction/results, and the methods make no mention of the requirements for replication. This is important because replication in an independent set helps alleviate potential concerns that association might be driven by (say) inflation in the initial Icelandic GWAS as replication in an independent set suggests the association is not a false positive.

- What is the threshold/requirement/definition for replication in UK Biobank? This should be in the methods
- The abstract etc. talks about UK Biobank as a replication set, and says variants were replicated line 33, 75, 186 etc but nowhere is what replication means defined e.g. what P-value threshold? Line 261 "We applied genome-wide-" seems to imply that the Bonferroni corrected genome-wide threshold is applied to both the Icelandic data and UK BB. Is that the case?
- Which model's p-value is used for replication, the additive or full model?
- Does the UK BB result need to be significant, or the meta-analysed result? If the latter that is not a

discovery/replication analysis, that is a meta-analysis (not suggesting which approach should be used, just noting it isn't clear).

- Why is the HLA association reaching $P = 0.01$ "nominal" significance (line 154)? What does nominal mean here? Doesn't that mean it wasn't replicated?

- Line 186 says "Access to a large foreign dataset allowed us to replicate the associations of all four variants with urticaria." That doesn't match up to line 154 where at least one result is nominal.

Which loci are formally significant in this study needs to be clearer; this overlaps with ambiguity about what replication means above. Results line 83 "In total, we found sequence variants at four loci that associate-" is ambiguous; only 2 loci look to be formally significant in the GWAS (Table 1); the other two while potentially interesting given their association with IgE don't look to pass the authors genome-wide significance threshold (e.g. $P < 7.5e-10$ for variants without predicted functional impact) so are at best promising until replicated in a larger hives GWAS. The relative strength/support for the 4 variants should be better reflected in the results, and the abstract when addressing the definition of replication here.

Why was transcriptomics performed in subcutaneous adipose? Line 124 and sup figure S10 notes GCSAML

was not detected in whole blood samples but why is adipose a more relevant than say mast cells, or other blood immune cells, or say even the skin where hives develop? Is it convenience (an accessible tissue that expresses the gene) or a relevant tissue for the functional role in hives? Discussion line 192 onwards suggests mast cells are the functional cell of interest, and subcutaneous adipose tissue is going to have few (if any) of the target mast cells as a proportion of the sequenced cells.

Related, the proteomics was in plasma – is that reasonable or plausible for genes downstream of GCSAML e.g. would the authors expect those genes (TPSAB1 etc) to be regulated by GCSAML, and then secreted into the plasma?

I note that the functional work is solid, interesting, and supports the lead variant impacting splicing and downstream protein expression; it is just that more discussion is needed to support that this tissue/observation is relevant to the disease itself, and discuss implications of using off target tissue.

Minor concerns

Abstract doesn't list P-values/confidence intervals, or even chromosomal positions for the associated variants. As such it is difficult to work out how many genome-wide significant loci there are – 2, 4?

Figure 2 – error bars should be defined in the figure legend.

Line 221 - Furthermore, our data are consistent with the causative nature of IgE in urticaria. This may need a caveat that the IgE derived variants are not formally significant?

Line 167 – "positive trend" has a P-value of 0.085 – and this should be stated more explicitly – was this significant at what threshold?

Throughout – using scientific notation for P values etc with values of 10^{-3} or more is potentially confusing e.g. Line 167, Table 1 – 0.01 is preferable to $1e-2$. 0.0021 to $2.1e-3$

Reviewers' comments:

Reviewer #1 (Remarks to the Author):

Sequence variant affects GCSAML splicing, mast cell specific proteins, and risk of urticaria

This paper face an unmet need in URTICARIA, it is the identification of some genetic hallmark that could help in the pathogenesis knowledge and further management of urticaria. The population studied is large and even two cohorts of people coming from two different countries were studied showing reproducible data. It's an interesting approach. There are some questions that should be taken into account directly related with the clinical knowledge and management of the urticaria. The authors are far from this clinical practice and the phenotype expression of the urticaria and the different trigger factors. Urticaria is managed as a unique disease in this paper and it is not. The entire paper should take into account this fact. In general the overview of urticaria is weak. It would be interesting to know what the authors think this new knowledge will impact in the general management of urticaria.

1. Urticaria is estimated to affect 15-25% of individuals

Please specify to which type of urticaria belong this percentage as is not the same when we refer to Acute Urticaria, to Chronic Urticaria, to Chronic Spontaneous Urticaria and to Chronic Inducible Urticaria .

In this case we were referring broadly to urticaria without consideration for sub-types. A recent review, assessed the prevalence of chronic urticaria, a sub-type of urticaria, to be 0.5% in Europeans and 1.4% in Asians. (**Added reference with** Pubmed ID 31494963).

We have accordingly changed the text to read (Page 2, line 6).

“Urticaria in any form is estimated to affect 15-25% of individuals, with chronic urticaria representing a rare form (0.5% in Europeans and 1.4% in Asians)”

2. WGS, please specify the meaning of this acronym in the first paragraph of the section “Results “

When first appearing in the manuscript, in the results section, we are now giving the full term and its acronym (Page 3, line 3). It now reads

“whole genome sequencing (WGS)”

3. Methods – Study Subjects and Genotyping

According with the authors, to identify the urticaria patients was used the (ICD10) code L50, this code include all types of urticaria independently is acute, chronic spontaneous urticaria or inducible.

L50 include, Allergic, cholinergic, chronic, due to food, due to heat, due to cold, due to plants, Dermographic, no specific, facticia (which is the same than Dermographic), idiopathic, non allergic, periodic recurrent, contact, solar, thermic (heat and cold), vibratory.

ICD was available but do not represent the actual classification of urticaria, but it is now a days the unique available until the ICD 11 will be implemented.

As L50 include different types of urticaria, acute and chronic, independently of the endogenous or exogenous trigger factor this should be well explained from the introduction. In the introduction of the paper the authors just mention the main and not unique mechanism described in Chronic Urticaria, spontaneous and inducible, the autoimmune hypothesis. This is mechanism is absolutely independent from the mechanism involved e.g. in the pure and minority allergic urticaria. The L50 ICD code, which is the source of this study, includes all classic phenotypes of acute and chronic urticaria. This is an important fact to be considered and should no be mixed as it will provide a confuse pathogenic approach to the different types of urticaria.

We thank you for the observation and do agree with you on the matter. It would be of interest to assess the effects of the identified variants on a well-characterized urticaria set to see if the effects differ on the different sub-types of urticaria. However, detailed sub-phenotypes and duration of symptoms were not available in this study. We therefore performed the association analysis on all urticaria cases, irrelevant to sub-phenotypes and duration of symptoms. Further analysis is up to later studies.

In accordance with the reviewer's comment, we have now clarified the ICD code we used in the study in the following places and now reads):

In the Introduction (Page 2, line 3)

“These outbreaks can occur as a reaction to allergens and physical stimuli or in an autoimmune or hereditary disorder, but the majority of outbreaks are idiopathic”

Page 2, line 15 of Introduction

“There is evidence that two mechanisms contribute to the pathogenesis of chronic urticaria...”

Page 2, line 27 of Introduction:

“To search for sequence variants affecting risk of urticaria, we performed a genome-wide association study (GWAS) on individuals with ICD10 code L50 and all sub-codes using data from Iceland...”

Page 3, line 1 of Results:

“We performed a GWAS on 14,312 Icelanders diagnosed with urticaria (ICD10 code L50 and all sub-codes) and 354,647 controls.”

Page 7, line 10 of Discussion:

“The ICD10 code used in this GWAS study, L50, includes various sub-types of urticaria depending on a defined trigger. These include allergic, physical, and idiopathic urticaria, both chronic and acute. Even if a trigger can be identified, a large majority of urticaria is idiopathic. Detailed sub-phenotypes and duration of symptoms were not available in this study and it would be of interest to assess the effects of the identified variants on a well characterized urticaria set to see if the effects differ on the different sub-types of urticaria.”

Page 9, line 8 of Methods:

“L50 covers various types of urticaria, i.e. allergic, physical, and idiopathic urticaria”

4. Methods - Association analysis

In this analysis, I understand that it was not considered the different types of urticaria available.

To identify urticaria cases, we searched for patients with International Classification of Diseases (ICD10) diagnosis code L50 and all subcodes in the Icelandic and UK population. We do not determine different sub-phenotypes or duration of symptoms in the study.

See also answer to comment #3

5. Please specify the donors signed which type of informed consent. If its a general informed consent for genetic studies, or specifically for this study

As approved by the National Bioethics Committee (approval no. VSN_19-157 or earlier VSN_14-099) all blood donors have either signed a specific consent for participation in this project or a broad consent allowing the use of samples and data derived from them for all deCODE studies approved by the National Bioethics Committee.

6. Results – GCSAML

The authors do an interesting exercise, looking for the differences between a single or recurrent episodes showing similar effects. And is good showing no differences according this course behavior. This approach does not take care of the true real life diagnosis o management of urticaria and its phenotypes. Although it was not the objective of this study this limitations should be taken into account.

As pointed by the reviewer, lifetime assessment of urticaria was neither available in Iceland nor in the UK. We have now added a paragraph to Discussions describing the sub-phenotyping of urticaria in our study.

Page 7, line 10 of discussion now says

“The ICD10 code used in this GWAS study, L50, includes various sub-types of urticaria depending on a defined trigger. These include allergic, physical, and idiopathic urticaria, both chronic and acute. Even if a trigger can be identified, a large majority of urticaria is idiopathic. Detailed sub-phenotypes and duration of symptoms were not available in this study and it would be of interest to assess the effects of the identified variants on a well-characterized urticaria set to see if the effects differ on the different sub-types of urticaria.”

7. GCSAML and blood cell traits.

The authors associate the risk-allele with a reduction in basophils, that was well described in chronic urticaria.

At this point also different considerations should be taken into account. Basopenia in urticaria is not a generalized finding in general practice and it depends about how basophils and which type. As it is mention it has been described mainly in chronic spontaneous urticaria. I this study this subtype analysis was not performed. A fact to be discussed.

See response to Comment 6

8. The author ignore in the paper previous Transcriptome studies done in urticaria as are the e.g. following:

Allergy Rhinol (Providence). 2015 Jan;6(2):101-10. doi: 10.2500/ar.2015.6.0124. Gene expression profiles in chronic idiopathic (spontaneous) urticaria. Patel OP1, Giorno RC, Dibbern DA, Andrews KY, Durairaj S, Dreskin SC.

Allergy. 2017 Nov;72(11):1778-1790. doi: 10.1111/all.13183. Epub 2017 May 26. Transcriptome analysis of severely active chronic spontaneous urticaria shows an overall immunological skin involvement. Giménez-Arnau A1, Curto-Barredo L1, Nonell L2, Puigdecanet E2, Yelamos J3, Gimeno R3, Rüberg S4, Santamaria-Babi L5, Pujol RM1

Allergy. 2019 Jan;74(1):141-151. doi: 10.1111/all.13547. Epub 2018 Oct 15. Omalizumab normalizes the gene expression signature of lesional skin in patients with chronic spontaneous urticaria: A randomized, double-blind, placebo-controlled study. Metz M1, Torene R2, Kaiser S3, Beste MT2, Staubach P4, Bauer A5, Brehler R6, Gericke J1, Letzkus M3, Hartmann N3, Erpenbeck VJ3, Maurer M1.

The suggested references have now been added to the text in Introduction chapter

Page 2, line 28 of Introduction now says

“Expression of GCSAML has hitherto not been associated with urticaria (ref).”

9. A final consideration about what is the impact of these findings in the Urticaria knowledge would be interesting to know.

We identify a splice donor variant in *GCSAML* that has a significant and substantial association with urticaria (particularly among homozygotes). Additionally, using transcript analysis we confirm that the variant affects splicing and through proteomics we identified three proteins for which the level in plasma associate with the disease allele. Taken together, these results improve our understanding of the etiology of urticaria. We demonstrate that *GCSAML* itself plays an important role in the pathogenesis of urticaria. We added a sentence in the discussion.

Page 8, line 17 of Discussion now reads

“The splice-donor variant is consistently the variant at this locus that most significantly associates with urticaria, basophil percentage, platelet count, RNA splicing of *GCSAML*, and levels of the three proteins. Together, these results demonstrate that *GCSAML* itself is important in the pathogenesis of urticaria.”

Reviewer #2 (Remarks to the Author):

General comments:

Authors reported GWAS of urticaria in an Iceland population. They reported a novel signal in GCSAML region, which was well replicated in independent cohort (UKBB). They added functional investigations by analyzing transcriptome- and proteome-wide analyses in the large Icelandic dataset; they showed an sQTL effect against GCSAML can explain the strongest GWAS association; and also showed the same variant is associated with plasma levels of tryptase, which is an important enzyme in angioedema. They also discussed signal similarity with GWAS of IgE. Overall, I appreciated the novelty and the importance of this study, and I believe this study improved our understanding of the etiology of urticaria. However, I think authors can improve the downstream analyses in several ways.

Major comments:

1. The effort to conduct a replication analysis in UKBB is a good strategy. However, since UKBB whole genetic information is now available, authors should be able to do GWAS of urticaria in UKBB and conduct meta-analysis for all variants (not limited to a few variants). In Table 1, I can see the meta-analysis statistics for only 4 variants. But I am curious to know the results of all other variants. I expect, they can find more novel signals from meta-analysis.

As this reviewer emphasized in his “General comments” chapter, we first used the Icelandic data as a discovery set. In order to replicate the finding in a different population we analysed the association in the UK Biobank.

We agree with the reviewer that both results from Iceland and UK can be combined in a larger meta-analysis, but then the UK Biobank cohort cannot be used for independent replication. We will therefore release summary statistics for all markers in Iceland for urticaria allowing other study groups to perform a meta-analysis using our data.

2. One key analysis missing in this manuscript is LD score regression analysis, which now becomes a standard approach in GWAS downstream analysis. It can be used in three purposes;

(i) Quality control of GWAS results: using intercept as an indicator of bias, authors should be able to provide how much of the genetic signals is bias (PMID: 25642630). It is a very important metric to evaluate the quality of GWAS results.

The reviewer is correct that the intercept is used to control for confounding. As described in the Supplementary material, we used linkage disequilibrium (LD) score regression to account for cryptic relatedness and population stratification. With a set of 1.1M variants, we regressed the χ^2 statistics from our GWA scan against LD score and used the intercept as a correction factor. *P*-values from the Icelandic and UK datasets were adjusted separately using LD score regression before being combined. The correction factors (intercepts) used are presented in Table S15.

Phenotype	Country	Correction factor Additive	Recessive
Urticaria (Total)	Iceland	1.097	1.046
	UK	1.021	1.016

(ii) Genetic correlations (PMID: 26414676). In Figure S17, they nicely showed the correlations of genetic effects between IgE and urticaria. This is very interesting and important to understand its etiology. However, they should extend this analysis to other diseases, not limited to IgE. Using genetic correlation based on LDSC, they should be able to assess genetic relationship of urticaria not only with IgE but also with a variety of diseases.

We emphasize that this analysis was not limited to IgE. Since asthma, atopic dermatitis, IgE, basophils, eosinophils, and platelets are linked to inflammation processes involved in the pathogenesis of urticaria, we tested 396 variants reported to associate with at least one of these traits for association with urticaria in Iceland and the UK (Table S1; Figures S17-S22).

As mentioned above, we are going to release the summary statistics for urticaria, and that would allow prospectively every researcher to check the genetic correlation between urticaria and a phenotype of their choice.

(iii) Partitioning heritability (PMID: 26414678). They can partition urticaria heritability using LDSC. If there are any publicly available epigenomic annotations of cell types which they believe important for urticaria (including mast cells), they can infer enrichment of heritability within such annotations. This analysis could provide genetic evidence that such cell types are important for urticaria.

As mentioned above, we are going to release the summary statistics for urticaria, which allows every researcher to determine the partitioning heritability from our data.

3. Related to sections of “Reported variants and urticaria”

- It is not clear how they selected “396 variants reported to associate with related immunological traits (L81)“ just by reading the main text. Are these 396 variants the all variants reported for these traits (e.g. in GWAS catalog)? This selection strategy is very critical since they set significant threshold at 0.05/396.

Yes, the reviewer is correct the 396 variants correspond to all significant variants reported for these traits in the GWAS catalog and we have made changes in the text to clarify this (See supplementary table 1).

Page 3, line 6of Results:

“... associations of 396 variants from the GWAS Catalog reported to associate with related immunological traits (Table S1) in the combined set of Iceland and UK ($P < 0.05/396$) and detected two other associations...”

- This is an analysis based on their hypothesis that variants associated with clinically related diseases (e.g. asthma) should also related to urticaria. This might be true, but they did not provide any genetic evidence to support their hypothesis. As I discussed above, they can add genetic correlation analysis to confirm their hypothesis: for example, they can ask their urticaria GWAS is significantly correlated with GWAS of asthma.

The hypothesis tested (e.g. for asthma) was: Do variants associating with asthma have an effect on urticaria proportional to their reported effect on asthma. The basis of the hypothesis is medical and biological and it turns out we do not see any support of this hypothesis to be correct.

As seen in Supplemenatry figure S19, the 30 sequence variants reported for asthma by Demenais and Pickrell, have an effect on urticaria that do not correlate with their asthma effect nor are they significantly associating with urticaria after accounting for multiple testing. Similar conclusion of absence of genetic correlation are reached for the other traits. Any type of genetic correlation analysis would largely be driven by the significant markers we are analysing. We are releasing summary statistics allowing other researchers to explore this in more detail.

- Also, they should not introduce signals detected in this analysis as significant associations in GWAS (as in abstract and L83). FCER1A and STAT6 should not be treated in the same way as GCSAML and HLA because the robustness of this analysis is much weaker than that of GWAS.

Through the abstract, results and tables we have each time clearly separated and stated that *GCSAML* and *HLA* signals come from a genome wide approach. In contrast, we emphasize that *FCER1A* and *STAT6* signals, which are genome wide significant for IgE, can only be detected to associate with urticaria when using a prior knowledge.

We have added text to the abstract and page 1, line 9 now reads (changes highlighted):

“Additionally, when assessing variants reported to associate with IgE levels, we found two variants, at the *FCER1A* and *STAT6* loci, associating with urticaria using a less stringent significance threshold”.

Minor comments:

- In abstract, authors utilized different significant thresholds. It is very confusing. First, they reported “We found two sequence variants associating with urticaria”. Next, they reported “we found two variants associating with urticaria”. Apparently, they utilized two different significant thresholds. They have to make abstract more readable on this point.

We have now changed the abstract to make this clearer:

Page 1 line 9

“We found two sequence variants associating with urticaria; the splice-donor variant rs56043070[A] in *GCSAML* and the intergenic variant rs28679786[T] at the HLA locus. Additionally, when assessing variants reported to associate with IgE levels, we found two variants, at the *FCER1A* and *STAT6* loci, associating with urticaria using a less stringent significance threshold.”

- Also in abstract, they utilized expressions like “variants in FCER1A” in several places. However, variants in gene does not make sense. They should say “variants in FCER1A region” or “variants in FCER1A loci”

We agree with the reviewer, we now have changed the text according to the suggestions.

Page 1 line 9

“at the *FCER1A* and *STAT6* loci”

- L148 “copy number” should be replaced with “allele count”. It is confusing with copy number variants.

We now have changed “copy number” to “allele count” (page 5 line 12)

- L153 “The association of this variant with urticaria replicates nominally in the UK (OR = 1.06, P-value = 0.01; Table 1).” They have to emphasize heterogeneity.

The p value for heterogeneity was already in the table and we are now adding to the text in the relevant paragraph

Page 5 line 17 now reads

“P-value = 0.01 and P-het = 0.02”

- Association pattern of rs56043070 is different in different traits. Some are well fit with additive model, others are not. Can they provide any reasonable speculations why this phenomenon was observed?

Overall, we observe an effect for homozygotes that is larger than expected by the additive effect for five out of six phenotypes where we observe an association under the full model. We are not willing to speculate why all patterns are not the same.

- Urticaria is a clinical diagnosis with multiple pathophysiology as they wrote in the introduction section. However, they did not clearly provide how they included patients with urticaria in the main text (L79). In addition to method, they should provide such critical information in the main text.

We have now added that the urticaria patients are identified through ICD10 code for urticaria- L50 and all sub-codes.

Page 2, line 25

“To search for sequence variants affecting risk of urticaria, we performed a genome-wide association study (GWAS) on individuals with ICD10 code L50 and all sub-codes using data from Iceland, and replicated the associations in data from the UK.”

Page 3, line 1 now reads:

“We performed a GWAS on 14,312 Icelanders diagnosed with urticaria (ICD10 code L50 and all sub-codes) and 354,647 controls.”

Reviewer #3 (Remarks to the Author):

This is a large GWAS of hives (urticaria) with an Icelandic discovery set and a UKBB derived replication set (total case N ~22,000, >750k controls). The Icelandic data set, as with other deCode analyses, was derived partially from sequencing and then imputation of untyped relatives. 4 loci were identified. These were followed up using transcriptomic and proteomics. The manner in which the trait was defined appeared appropriate. The results are likely to be of interest to people working in immunity/allergic disorders.

Major concerns

1a) No ancestry/population sub structure correction in the logistic regression GWAS performed in the Icelandic GWAS. The potential for inflation of test statistics (e.g. LDscore intercept) need to be described and discussed in the main text, not in the final paragraph of sup page 3, and in sup table 15. Sup table 15 shows the corrective factor (LD score intercept) for the main Urticaria GWAS is very high for the Icelandic set-1.10 - and should be addressed e.g. with principal components, and/or approaches that better handle population structure for binary traits such as SAIGE PMID: 30104761, not just corrected for without further investigation. The UK Biobank dataset does not seem to have the same issue for Urticaria GWAS (~1.03).

Does the hives association for the lead 4 SNPs remain if population structure is corrected for in the Icelandic GWAS? I suspect they will still be associated, but this should be confirmed. If the authors cannot use SAIGE etc for binary GWAS it is possible

to approximate a log(OR) from the bolt-lmm regression output e.g. see section 10.2 of the bolt-lmm guide at <https://data.broadinstitute.org/alkesgroup/BOLT-LMM/>.

We are confident in our results and that the probability of false positive results are unlikely to arise from population sub structure for two main reasons. Firstly, we note that geographic regions in Iceland (county of origin) are already accounted for in the logistic regression analysis in Supplementary methods/Association analysis, and we have previously performed and described it in many published articles. In Iceland, we had previously performed principal component analysis and describe that the county of origin is in concordance with the first principal components (PMID: 19503599). For this reason and that we perform family imputation based on long-range phasing and genealogy data, we use county of origin as a proxy covariate for the first principal components in our analysis. Secondly, studies in Iceland have shown that even if the population is genetically homogeneous, there is still an observable, but small, effect from population stratification. However, to address the population sub structure, we use LD score regression and use the intercept from the analysis as a correction factor.

We have added from the supplementary information to main methods the following

“We used linkage disequilibrium (LD) score regression to account for cryptic relatedness and population stratification in the data.”

We emphasize that correction factors are larger in related populations. In the case of Iceland (population of 340K inhabitants), we use over 60 % of the adults. The correction factors in Iceland are consistent with the high level of relatedness. In the UK, the relatedness of individual is minimal compared to Iceland.

1b) Bolt-LMM was used for quantitative traits which should hopefully be addressing any sub structure for the quantitative measures; for BOLT-LMM analyses it is more informative to report the attenuation ratio rather than LDSC intercept (since this can be inflated in mixed models; see PMID: 29892013). Ideally here the attenuation ratio should be < 0.08, and if not this should be discussed/addressed.

An attenuation ratio, which is $(\text{LDSC intercept} - 1) / (\text{mean } \chi^2 - 1)$, close to zero would indicate that inflation in the test statistics is to most extent due to polygenicity and not population stratification. For biobank scale data, where there are large/huge sample sizes and traits are very heritable this is to be expected. For Urticaria, where we do not have a very large sample size and the phenotype is not very polygenic the attenuation ratio is not expected to be <0.08. There is population structure in the data (more in Iceland than in the UKBB) that is sufficiently being accounted for by using the LDSC intercept to scale the test statistics. For Iceland the attenuation ratio is 0.71 and for UKBB it is 0.53.

We have added from the supplementary information to main methods the following

“We used linkage disequilibrium (LD) score regression to account for cryptic relatedness and population stratification in the data.”

2) The authors have used two models to test for association – additive and a full genotypic model. The full details for these models are in the supplemental, and that is probably okay given their length, but this should be referenced to in the main text method section.

In the Statistical Analysis chapter from the main text, we have now added a reference to the Supplementary method where we describe in details the two models we tested.

Page 9 line 15 now reads:

“We used two different models for association testing; the additive and the full genotypic model. For the GWAS the initial discovery was done in Iceland under the additive model (See Supplementary Methods)“.

3a)The approach for setting genome-wide significant threshold (page 9 line 261) in the discovery set is appropriate but should be described in the main methods not in the sup. Does this threshold apply just to the Icelandic GWAS data or UKBB as well? See following as well

We have now added a sentence in the main text (Statistical Analysis section) summarizing and referring to the Significance thresholds chapter in Supplementary Methods:

Page 10 line 3 now reads:

“We applied genome-wide significance thresholds corrected for multiple testing using adjusted Bonferroni procedure weighted for variant classes and predicted functional impact as we previously described (See supplementary methods).”

This threshold applied to the Icelandic discovery set only in the Genome wide association part

3c) It is not clear which association model's P-value is used to identify loci that reach genome-wide significance in the discovery set – additive, full etc.

In the GWAS, the discovery were made in Iceland using an additive model. In order to clarify the analysis we have modified the text in the Results and the Methods:

The Second sentence of the results, page 3 line 2, now reads

“The GWAS was performed using an additive model with 32.5 million variants identified through whole genome sequencing (WGS) of 28,075 Icelanders

In the Statitiscal Analysis chapter from the main text, we have added two sentences (Also see comment 2), page 9 line 14, that reads as follow:

“We used two different models for association testing; the additive and the full genotypic model. For the GWAS the initial discovery was done in Iceland under the additive model (See Supplementary Methods)“.

3d) What replication means in this article is very unclear; as a result I have a series of sub questions related to this. Overall the presentation of results for replication is hard to match up with the abstract/introduction/results, and the methods make no mention of the requirements for replication. This is important because replication in and independent set helps alleviate potential concerns that association might be driven by (say) inflation in the initial Icelandic GWAS as replication in an independent set suggests the association is not a false positive.

- What is the threshold/requirement/definition for replication in UK Biobank? This should be in the methods

We are in agreement with the reviewer concerning the importance of replication.

In the manuscript, we performed two types of analyses.

A) A genome wide analysis were we require genome wide significance in the discovery set and we then attempted replication in UK. Given that in this analysis we discovered two markers (one at GCSAML and one at HLA region), the replication threshold is $0.05/2=0.025$. Both variants that reached genome signifance in Iceland were replicated at 0.025 level in UK.

B) Additionally, we assessed urticaria associations of 396 variants reported to associate with related immunological traits in the combined set of Iceland and UK.

To clarify, first paragraph of the results now reads:

“In Iceland we detected two genome wide significant associations at *GCSAML* and *HLA*, which were then replicated in the UK using a set of 8,730 urticaria cases and 399,923 controls (Both $P < 0.05/2$). Additionally, we assessed urticaria associations of 396 variants from the GWAS Catalog reported to associate with related immunological traits

(Table S1) in the combined set of Iceland and UK ($P < 0.05/396$) and detected two other associations with variants reported to associate with IgE level (at *FCERIA* and *STAT6*). In total, we found sequence variants at four loci that associate with urticaria; *GCSAML*, *HLA*, *FCERIA*, and *STAT6* (Tables 1 & S2, Figure S1).”

We further supported our findings by looking up association results for these four markers in Finland using 2,926 urticaria cases and 131,988 controls (FINNGEN project, http://r3.finngen.fi/pheno/L12_URTICARIA) and in Japan using 9,893 cases and 162,190 controls (PMID 34594039). Three of the associations could be replicated, of which the association represented by a variant in *GCSAML* associated with urticaria has already been reported in a study from Japan.

- The abstract etc. talks about UK Biobank as a replication set, and says variants were replicated line 33, 75, 186 etc but nowhere is what replication means defined e.g. what P-value threshold? Line 261 “We applied genome-wide-“ seems to imply that the Bonferroni corrected genome-wide threshold is applied to both the Icelandic data and UK BB. Is that the case?

For the Icelandic dataset we determined genome-wide significance thresholds using adjusted Benferroni procedure weighted for variant classes.

Since we tested two variants in the replication set, i.e. UK BioBank, the P value threshold will be $0.05/2 = 0.025$.

We have added a text to the first paragraph of the results, which now reads (page 3, line 4):

“...which were then replicated in the UK using a set of 8,730 urticaria cases and 399,923 controls (Both $P < 0.05/2$).”

- Which model’s p-value is used for replication, the additive or full model?

We used the additive model to determine replication to urticaria, not the full model. We note that the p value are significant for both models.

- Does the UK BB result need to be significant, or the meta-analysed result? If the latter that is not a discovery/replication analysis, that is a meta-analysis (not suggesting which approach should be used, just noting it isn’t clear).

In the genome wide association we require genome wide significance in the discovery set in Iceland and we then attempted replication in UK. Given that in this analysis we discovered two markers (one at *GCSAML* and one at *HLA* region), the replication threshold is $0.05/2=0.025$. Both variants that reached genome-wide significance in Iceland were

replicated at 0.025 level in UK. However, during the review process, the association of a sequence variant in *GCSAML* with urticaria has been reported.

Additionally, we assessed 396 variants reported to associate with related immunological traits in the combined set of Iceland and UK. Using this prior knowledge, two of these survive multiple testing (0.05/396) in the combined set of Iceland and UK. Of these two, one would also survive that many test in the Icelandic group but the other does not. Since submission, we searched for further follow up dataset, and we accessed association available online in a Finnish set (FINNGEN) and later in a Japanese set (Biobank Japan). In these datasets we could replicate all associations we reported but the one represented by a variant in the HLA region of chromosome 6 (not tested).

	ClosestGene	japan		finland	
		P	or	P	or
rs56043070	GCSAML	6,9E-12	1,24	3,2E-04	1,55
rs28679786	[HLA]	-	-	-	-
rs2251746	FCER1A	4,6E-04	0,90	2,5E-03	0,93
rs1059513	[STAT6]	0,049	0,94	8,4E-04	0,85

- Why is the HLA association reaching P = 0.01 “nominal” significance (line 154)? What does nominal mean here? Doesn’t that mean it wasn’t replicated?

You are correct. We have now removed the word “nominally” from the sentence.

- Line 186 says “Access to a large foreign dataset allowed us to replicate the associations of all four variants with urticaria.” That doesn’t match up to line 154 where at least one result is nominal.

This marker was initially genome wide significant in the discovery set in Iceland and replicates in the UK Biobank after adjusting for multiple testing. We have removed the word “nominally”.

We have now changed the sentence and it reads “The association of this variant with urticaria replicates in the UK (OR = 1.06, P-value = 0.01; Table 1).”

Which loci are formally significant in this study needs to be clearer; this overlaps with ambiguity about what replication means above. Results line 83 “In total, we found sequence variants at four loci that associate-” is ambiguous; only 2 loci look to be formally significant in the GWAS (Table 1); the other two while potentially interesting given their association with IgE don’t look to pass the authors genome-wide significance

threshold (e.g. $P < 7.5 \times 10^{-10}$ for variants without predicted functional impact) so are at best promising until replicated in a larger GWAS. The relative strength/support for the 4 variants should be better reflected in the results, and the abstract when addressing the definition of replication here.

Through the abstract, results and tables we have each time clearly separated and stated that GCSAML and HLA signal come from a genome wide approach. In contrast we emphasize that FCER1A and STAT6 signals, which are genome wide significant for IgE, can only be detected to associate with urticaria when using a prior knowledge.

To clarify page 1 line 9 now reads :

“Additionally, when assessing variants reported to associate with IgE levels, we found two variants, at the FCER1A and STAT6 loci, associating with urticaria using a less stringent significance threshold.”

Since the submission of the manuscript Fingen and Biobank Japan have released summary level data for urticaria. The FCER1A and STAT6 associations do replicate in both datasets. In a study of 220 phenotypes in Japan recently published the association of the splice donor variant in *GCSAML* and urticaria is reported. The study also reported two other associations of variants at STIM1 and TPSB2, which we replicate.

	ClosestGene	japan		finland	
		P	or	P	or
rs56043070	GCSAML	6,9E-12	1,24	3,2E-04	1,55
rs28679786	[HLA]	-	-	-	-
rs2251746	FCER1A	4,6E-04	0,90	2,5E-03	0,93
rs1059513	[STAT6]	0,049	0,94	8,4E-04	0,85

The scope of this paper was mainly to describe the GCSAML association with urticaria. A large meta-analysis follow-up study could therefore be conducted, as we will release our summary level data.

4) Why was transcriptomics performed in subcutaneous adipose?

Line 124 and sup figure S10 notes GCSAML was not detected in whole blood samples but why is adipose a more relevant than say mast cells, or other blood immune cells, or say even the skin where hives develop? Is it convenience (an accessible tissue that expresses the gene) or a relevant tissue for the functional role in hives? Discussion line 192 onwards suggests mast cells are the functional cell of interest, and subcutaneous adipose tissue is going to have few (if any) of the target mast cells as a proportion of the sequenced cells.

We used RNA sequencing to assess the impact of the GCSAML splice donor sequence variant on RNA splicing. We have previously performed RNA sequencing from whole blood and from adipose tissue in Icelanders. We did not have any RNA sequencing from mast cells or skin. Expression of *GCSAML* in whole blood was too low for analysis. We then focused on the adipose tissue samples (n=750) that were available. This analysis allowed us to conclude that the splice donor leads to a shift in transcript usage.

Page 4, line 19 now reads:

“The impact of the splice-donor variant rs56043070[A] on RNA-splicing was analyzed in RNA-Seq data from subcutaneous adipose tissue in 750 Icelanders since the data were available to us. Also, because the expression of GCSAML in whole blood was too low for further analysis (Figure S10).”

5) Related, the proteomics was in plasma – is that reasonable or plausible for genes downstream of GCSAML e.g. would the authors expect those genes (TPSAB1 etc) to be regulated by GCSAML, and then secreted into the plasma?

We have previously performed a Proteomics assay from plasma in a set of close to 40K Icelanders.

We found that plasma levels of the TPSAB1 and TPSB2 and KIT protein-products increase with rs56043070[A] copy number. For two of these proteins the results were robust to stratify on disease status (similar effect of the variant on the protein in cases and controls Figure S16), indicating that the observed increased plasma protein levels are driven by genotype independently of disease status. We are not willing to speculate on the exact mechanism by which the protein levels are influenced. We mentioned that all three proteins are linked to mast cell activation, which is at the center of the pathogenesis of urticaria. We also mentioned that FANTOM5 dataset identified *GCSAML*, *FCERIA*, *TPSAB1*, *TPSB2*, and *KIT* as either exclusively or preferentially expressed in mast cells³⁸. The association profile of this variant, including all protein products significantly affected by it, indicates that *GCSAML* plays a role in mast cell regulation, with the splice-donor variant leading to greater mast cell activation.

There is a great variety of expression patterns in different tissues. The plasma proteome consist of proteins expressed in various tissues that are secreted into the plasma and from cell death. The observed association of rs56043070[A] with protein levels in our results is likely

happening through effects in mast cells and the proteins are released into the plasma. Studies of the secretome have demonstrated that the protein products of *TPSAB1* and *TPSB2* are secreted into the plasma.

As a conclusion, we do think that the genes encoding the proteins we see associating with the splice-donor variant in *GCSAML* are likely regulated by *GCSAML* and then secreted into the plasma.

6) I note that the functional work is solid, interesting, and supports the lead variant impacting splicing and downstream protein expression; it is just that more discussion is needed to support that this tissue/observation is relevant to the disease itself, and discuss implications of using off target tissue.

We did not have access to a more relevant tissue, like skin, or cell lineages, like mast cells. We have blood samples from 13,175 individuals and adipose samples from 750 individuals available for RNA analysis. Expression in blood was not high enough to be used, but we could use adipose tissue. Doing the analysis on one tissue gives limited information on RNA splicing in other tissues, due to large diversity of splicing between tissues. However, our data shows the variant does affect splicing in one tissue and is therefore an indication of what we might see in other tissues.

To address this, we have added the following to Discussions (line 27, page 7):

We note that the effects of rs56043070[A] on RNA splicing in adipose tissue gives limited information on splicing in a tissue more relevant to urticaria, such as skin, or cell lineages, such as mast cells. The relevance in urticaria pathology is therefore not obvious. However, our data show that the splice-donor variant is capable of altering the splicing of *GCSAML* transcripts and it would be of interest to study this further in more relevant tissues or cell lineages to assess the relevance on urticaria pathology.

Proteins in plasma come either from cells in various tissues that secreted them or from cell death. The association between rs56043070[A] and protein levels of three proteins in plasma might therefore be a sign of what is happening in other tissues of the body, such as the skin. The two tryptases expressed from *TPSAB1* and *TPSB2* that are enriched for in mast cells have been reported to be secreted to plasma. We did not have access to more relevant tissue like mast cells but expect similar results there.

Minor concerns

7) Abstract doesn't list P-values/confidence intervals, or even chromosomal positions for the associated variants. As such it is difficult to work out how many genome-wide significant loci there are – 2, 4?

We have now added P values, effect and frequency for the key results on urticaria association. In brief we a) discover two genome significant and replicate them in UK b) detect association with urticaria of two reported variants with a reported association to related traits ($p < 0.05/396$) in Iceland and UK .

We changed the abstract accordingly

Page 1, line 5 reads:

“We found two sequence variants associating with urticaria: the splice-donor variant rs56043070[A] (Hg38: chr1:247556467) in *GCSAML* (MAF = 6.6%, OR = 1.25 (95% CI: 1.18-1.33), P-value = 3×10^{-13}) and the intergenic variant rs28679786[T] (Hg38: chr6:32558495) at the HLA locus (MAF = 25.5%, OR = 1.13 (95% CI: 1.09-1.17), P-value = 4×10^{-11}). Additionally, when assessing variants reported to associate with IgE levels, we found two variants, in *FCER1A* and *STAT6*, associating with urticaria using a less stringent significance threshold. “

8) Figure 2 – error bars should be defined in the figure legend.

We have now added the definition in the figure legend

The legend now reads):

“Boxplot of the quantified proportions of two GCSAML transcripts; the primary transcript (ENST00000366488.5) and the novel transcript splicing out exon 2, stratified by rs56043070 genotype. After removing 78 samples due to lack of expression of the first three exons, we used 672 samples for the analysis (Supplementary Methods). The hinges of the boxes represent the quartiles of the distribution, and whiskers represent the highest and lowest point no further than 1.5 times the interquartile range from the upper and lower hinges, respectively. For more details, see Figures S10-S15.”

9) Line 221 - Furthermore, our data are consistent with the causative nature of IgE in urticaria. This may need a caveat that the IgE derived variants are not formally significant?

This sentence in the last part of the discussion, is emphasizing that our data are suggestive of a relationship of the effects of IgE variants on IgE and Urticaria. We are saying that this relationship is consistent with previous discussion as mentioned in the introduction.

Also, the further confirmation in Finland and Japan of association of IgE variants at *STAT6* and *FCERIA* with urticaria should increase the confidence in these observations.

10)Line 167 – “positive trend” has a P-value of 0.085 and this should be stated more explicitly – was this significant at what threshold?

Thank you for pointing this out. We have now fixed it and the text now says (Page6, line 3):

“We do not observe a correlation between the reported effects of variants on IgE levels and their effects on urticaria (regression slope: 1.53, P-value: 0.085).”

11) Throughout – using scientific notation for P values etc with values of 10⁻³ or more is potentially confusing e.g. Line 167, Table 1 – 0.01 is preferable to 1e-2. 0.0021 to 2.1e-3

Thank you for the suggestion. We have now made changes according to the suggestion

Reviewer #3 (Remarks to the Author):

Response (1a) - minor comment only

Thanks for the comprehensive and clear answer. I am satisfied with the content of the response, but this information should all be readily accessible in this manuscript rather than in other publications. In addition to the added text I would (briefly) summarise the points the authors provided in this response in the main text method's e.g. that a covariate is fitted that proxies principle components and cite PMID 19503599.

Related, the inserted text "We used linkage disequilibrium (LD) score regression to account for cryptic relatedness and population stratification in the data-" could be much clearer. E.g. If this means the SEs etc. were adjusted by the intercept this should be stated plainly. That is, describe exactly what was done to use LDSC to account for cryptic relatedness etc.

The authors' response has addressed my other raised concerns to my satisfaction.

Reviewers' comments:

Reviewer #2 (Remarks to the Author):

Summary:

The authors added some requested analyses, which are satisfactory. However, they did not conduct other analyses for unclear reasons. Again, I appreciate the novelty and the importance of this study, and I believe this study improved our understanding of the etiology of urticaria. However, there are remaining issues to be addressed.

Major comments:

1, the LDSC intercept is very high (1.097). As shown in the original article (PMID: 25642630), the intercept is an index of the bias in the GWAS sumstats. They used it as a correction factor, but this is far from sufficient. They need to explain why so much bias remained in this GWAS, what effort they conducted to avoid the bias. Did they try GLMM (such as SAIGE)? At least, they need to provide a detailed discussion of why they believe their analytic strategy is robust.

The LDSC intercept was 1.02 for the UK Biobank and 1.097 for the Icelandic population. We emphasize that correction factors are larger in more related populations. In the case of Iceland (population of 340K inhabitants), we use over 60 % of the adults and correction factors in Iceland are consistent with the level of relatedness. In the UK Biobank, the relatedness of individuals is minimal compared to Iceland. In Loh et al., 2018 (PMID:29892013), Supplementary Table 4(a) shows LDSC intercepts for various phenotypes in the UK biobank database and for 14 of 23 phenotypes the LDSC intercept is equal or more than 1.097 when using BOLT with principal components (PC). An intercept of 1.097 is therefore not unexpected and not surprisingly high as the reviewer states. The LD score method is meant to distinguish between inflation from a true polygenic signal and bias and using the intercept as a correction factor corrects for the bias.

In our logistic regression analysis, we additionally account for geographic regions in Iceland (county of origin) and have described it in many published articles. In a principal component analysis in Iceland, we have described that the county of birth is in concordance with the first principal components (PMID: 19503599). We are confident that we have sufficiently adjusted for population structure and replication in the UK Biobank confirms that the identified associations are not false positives.

We have now added a sentence in Methods saying, lines 278-282, page 10:

“To account for cryptic relatedness and population stratification in the data, we adjusted the χ^2 statistic from the logistic regression by dividing it with the intercept (correction factor) from linkage disequilibrium (LD) score regression analysis. The correction factors that we obtained are consistent with the ones reported by others (PMID:29892013).”

2, It seems that the authors do not understand genetic correlation analysis (PMID: 26414676), which I introduced in the initial revision. This is a polygenic analysis to test the genetic relationship between diseases and traits. This method does not solely rely on the variants with significant associations (as they did in their manuscript). Therefore, their statement in the response letter (“Any type of genetic correlation analysis would largely be driven by the significant markers we are analysing”) is not correct and is not a reasonable excuse for not doing genetic correlation analysis.

Thank you for the reiteration regarding genetic correlation. We have now performed a genetic correlation analysis. We used a cross-trait LD score regression and summary statistics for Urticaria L50 diagnoses in the deCODE dataset and from the UKB dataset for other traits. In these analyses, we used results for about 1.2 million imputed variants, and for LD information we used precomputed LD scores for European populations. The results of genetic correlation can now be seen in Tables S14 and S15. The genetic correlation analysis requires summary level data for the traits that are analysed and are not necessarily available for all traits of interest. E.g., there was no available GWAS for IgE and IgE is not one of the traits systematically assessed in the UKB. After correcting for multiple testing, we observed a genetic correlation between urticaria and asthma but not with any other diseases or traits.

We have added a sentence in Results. Lines 172-176, Page 6 now says:

"We explored the genetic correlation between urticaria (ICD10 code L50) diagnosis and 1,985 case control phenotypes defined from ICD codes and 17 quantitative phenotypes in the UKB dataset. After correcting for multiple testing (significance thresholds: $P < 0.05/2002 = 2.5 \times 10^{-5}$), we found a genetic correlation between urticaria (ICD10 code L50) and asthma diagnoses (ICD10 code J45) ($r_g = 0.38$, $P = 1.2 \times 10^{-5}$) (Table S14 & S15) but not with any other diseases or traits. "

Furthermore, we have added a text to the statistical analysis subsection in the Methods section for clarification. Lines 286-290, Page 10 now says:

"We calculated genetic correlations between urticaria (ICD10 code L50) diagnoses and other traits as follows: We used cross-trait LD score regression and summary statistics for urticaria (ICD10 code L50) in the deCODE dataset and from the UKB dataset for other traits. In these analyses, we used results for about 1.2 million well imputed variants, and for LD information we used precomputed LD scores for European populations (downloaded from: https://data.broadinstitute.org/alkesgroup/LDSCORE/eur_w_ld_chr.tar.bz2)."

3, I mentioned in the initial comments “This is an analysis based on their hypothesis that variants associated with clinically related diseases (e.g. asthma) should also be related to urticaria.”, referring to the analysis where they used the significant threshold of 0.05/396.

They wrote in the response letter that “The hypothesis tested (e.g. for asthma) was: Do variants associating with asthma have an effect on urticaria proportional to their reported effect on asthma. The basis of the hypothesis is medical and biological and it turns out we do not see any support of this hypothesis to be correct.” Having seen these negative results, do they still believe their strategy is robust? They need to provide valid explanations why this strategy (restricting the analysis to the related disease-associated variant and loosening the significant threshold) is an acceptable strategy.

We explored the genetic relationship between urticaria and various phenotypes, both by comparing the effects of reported variants to their effects on urticaria and now by performing genetic correlations as suggested by the reviewer. In the genetic correlation analysis, we observed a significant genetic correlation between urticaria and asthma ($r_g = 0.38$, $P = 1.2 \times 10^{-5}$; see answer to comment #2). We were not able to explore genetic correlation between IgE and urticaria using LDSC regression since summary level data from GWAS studies on IgE are not available and IgE is not measured routinely in the UK Biobank.

Chronic urticaria is associated with development of autoantibodies to IgE and is therefore of interest to explore the few reported variants in more detail (PMID: 30984191). We found two variants, at the *FCERIA* and *STAT6* loci, associating with urticaria using a less stringent significance threshold. These two results did not reach genome-wide significant in our initial analysis and these results were separated from the main findings.

Since our submission other datasets have been made publicly available giving the opportunity to further assess our results at *FCERIA* and *STAT6*. Using those, the variant in *FCERIA* is genome-wide significant in the combination of our data and publicly available data from Finland and Japan ($P\text{-value} = 2.7 \times 10^{-15}$) and replicates with similar effects in each of these two groups. The variant in *STAT6* also replicates in each of these two groups and the combined association becomes more significant ($P\text{-value} = 6.7 \times 10^{-9}$). The replication of these variants and the genetic correlation with asthma supports the initial hypotheses to scrutinize in this urticaria GWAS variants reported for asthma and IgE.

We have added Table S13 showing the comparison of the associations of these two variants with urticaria in Iceland, UK, Finland, and Japan.

Table S13. Comparison of associations of variants at *FCER1A* and *STAT6* with urticaria diagnosis in Iceland, UK, Finland, and Japan. Meta analysis of data from Finland-Japan and all cohorts is shown as well. OR = Odds ratio

rs name	Closest Gene	Iceland - aster		UK		Finland		Japan		Finland-Japan meta		all meta	
		P	OR	P	OR	P	OR	P	OR	P	OR	P	OR
rs2251746	FCER1A	1.1E-06	0.92	2.1E-03	0.95	2.4E-06	0.92	4.6E-04	0.90	5.2E-09	0.92	2.7E-15	0.93
rs1059513	[STAT6]	2.2E-03	0.92	1.6E-04	0.91	7.9E-03	0.90	0.049	0.94	1.41E-03	0.92	6.7E-09	0.92

We have added a sentence in Results. Lines 167-170, Page 6 now says:

“The associations of the two variants at *FCER1A* and *STAT6* with urticaria also replicate with similar effects in recent publicly available data from Finland and Japan, reaching in the combined set a P-value of 2.7×10^{-15} and 6.7×10^{-9} , respectively (Table S13).”

We added a footnote in Table 1 referring to Table S13, line 335, Page 13:

“*For further replication in Japan and Finland see table S13”

Reviewer #3 (Remarks to the Author):

Response (1a) - minor comment only

Thanks for the comprehensive and clear answer. I am satisfied with the content of the response, but this information should all be readily accessible in this manuscript rather than in other publications. In addition to the added text I would (briefly) summarise the points the authors provided in this response in the main text method's e.g. that a covariate is fitted that proxies principle components and cite PMID 19503599.

We have now added a sentence to the methods section. Lines 270-272, Page 9 now say:

“We used county of birth as a proxy covariate for the first principal components in our analysis because county of birth has been shown to be in concordance with the first principal components in Iceland (PMID: 19503599).”

Related, the inserted text “We used linkage disequilibrium (LD) score regression to account for cryptic relatedness and population stratification in the data-” could be much clearer. E.g. If this means the SEs etc. were adjusted by the intercept this should be stated plainly. That is, describe exactly what was done to use LDSC to account for cryptic relatedness etc.

We have made changes accordingly and, lines 278-282, page 10, now says:

“To account for cryptic relatedness and population stratification in the data, we adjusted the χ^2 statistic from the logistic regression by dividing it with the intercept (correction factor) from linkage disequilibrium (LD) score regression analysis. The correction factors that we obtained are consistent with the ones reported by others (PMID:29892013) (Table S18)”.

The authors' response has addressed my other raised concerns to my satisfaction.

Reviewers' comments:

Reviewer #2 (Remarks to the Author):

Summary:

The authors added several analyses and discussions in the revised manuscript. However, I do not think they have fully addressed my concerns. The most important message to the authors is that they are sticking to their initial analysis strategy and won't adjust their strategy even when they observed inconsistent results that do not support their intention. This is a major flaw in their scientific discussion. Please see my comments below.

Minor comments:

1, I appreciate their additional analyses of genetic correlation. To summarize the author's findings:

- Among traits except for IgE, they observed significant genetic correlation only with asthma.
- For IgE level, the authors wrote in the main text "we do not observe a correlation between the reported effects of variants on IgE levels and their effects on urticaria (regression slope: 1.53, P-value: 0.085)".

I understand that asthma, atopic dermatitis, IgE, basophils, eosinophils, and platelets are linked to inflammation processes involved in the pathogenesis of urticaria. However, this medical (or biological) prior knowledge can't be used as a basis for restricting the variant set (396 variants) for the downstream analyses without further discussion. So far, they only validated the use of asthma-associated variants (30 variants they listed in Supplemental table 1).

In light of these facts, I suggest the authors change their analytical strategy. Currently,

- i) they first arbitrarily restricted the variant set using their hypothesis (which they failed to validate)
- ii) they next identified some associations with a permissive threshold
- iii) they finally validated such associations using independent GWAS results.

My suggestion is as follows:

- i) first conduct genome-wide meta-analysis using their sumstats and other urticaria GWAS they used in Table S13.
- ii) they should be able to detect some novel associations including FCER1A and STAT6 (they can keep all of the discussion regarding FCER1A and STAT6 since both were significant after meta-analysis as they showed)
- iii) they can test the overlap of urticaria-associated variants with IgE-associated ones.

This will produce a more robust discussion. I understand that the two urticaria-associated variants at FCER1A and STAT6 were significant after meta-analyzing their results with other recent GWAS.

Therefore, I don't think these associations are false positives. My point is that even if they report the same associations (FCER1A and STAT6), the reasonable process of reaching the conclusion is vital in scientific articles.

2, The authors need to interpret the intercept very carefully. In Loh et al (PMID:29892013; the article they cited in the response letter), Loh explained the tricky aspect of the intercept in the main text as follows: "we observed that while the value of the LD score regression intercept (previously proposed as an indicator of confounding) was generally difficult to interpret due to attenuation bias, which causes the intercept to rise above 1 with increased sample size and heritability". So, the paper they cited was claiming that we need to interpret intercepts carefully due to the attenuation bias (see the definition in Loh et al). In GWAS results with a large sample size (such as QTL-GWAS reported in Loh et al), we usually observe increased mean chi-squared, which leads to increased intercept (the intercept value in large-scale GWAS tends to be much larger than those in moderate-power GWAS). Therefore, the intercept values in Loh et al can never be fairly compared with this GWAS's intercept. How was the attenuation ratio, which was described in Loh et al. The authors can fairly compare their attenuation ratio with those reported in Loh et al (the attenuation ratio is not influenced by the scale

of GWAS). If the attenuation ratio is very high compared with the standard value (as reported in Loh et al), this fact should be discussed in the main text as a potential limitation of this GWAS.

Again, the use of intercept as a correction factor of GWAS is just one remedy for the bias in this GWAS. But, of course, this strategy is not at all perfect. Explaining to the readers the potential bias in the GWAS is crucial in scientific articles.

Reviewers' comments:

Reviewer #2 (Remarks to the Author):

Summary:

The authors added several analyses and discussions in the revised manuscript. However, I do not think they have fully addressed my concerns. The most important message to the authors is that they are sticking to their initial analysis strategy and won't adjust their strategy even when they observed inconsistent results that do not support their intention. This is a major flaw in their scientific discussion. Please see my comments below.

Minor comments:

1, I appreciate their additional analyses of genetic correlation. To summarize the author's findings:

- Among traits except for IgE, they observed significant genetic correction only with asthma.**
- For IgE level, the authors wrote in the main text "we do not observe a correlation between the reported effects of variants on IgE levels and their effects on urticaria (regression slope: 1.53, P-value: 0.085)".**

I understand that asthma, atopic dermatitis, IgE, basophils, eosinophils, and platelets are linked to inflammation processes involved in the pathogenesis of urticaria. However, this medical (or biological) prior knowledge can't be used as a basis for restricting the variant set (396 variants) for the downstream analyses without further discussion. So far, they only validated the use of asthma-associated variants (30 variants they listed in Supplemental table 1).

In light of these facts, I suggest the authors change their analytical strategy. Currently,

i) they first arbitrarily restricted the variant set using their hypothesis (which they failed to validate)

ii) they next identified some associations with a permissive threshold

iii) they finally validated such associations using independent GWAS results.

We thank the reviewer for their comments and suggestions and we have modified the manuscript following all the suggestions. Initially, the discovery part of the study was a GWAS of urticaria in Iceland, where we used the UK Biobank data for replication. Now the study is a meta-analysis of GWAS of urticaria from Iceland, UK, Finland, and Japan as suggested by the reviewer. The association we reported at the *GCSAML* locus still represents the most significant association with the greatest effect size with urticaria in the updated analysis. The associations with the variants at *FCER1A* and *STAT6* now reach genome-wide significance. In the meta-analysis we detect associations with urticaria risk with nine variants in total. The genes these sequence variants are in point at the importance of mast cells and type 2 immune response in the pathophysiology of urticaria.

My suggestion is as follows:

i) first conduct genome-wide meta-analysis using their sumstats and other urticaria GWAS they used in Table S13.

As the reviewer suggested, we have now redone the study as a genome-wide meta-analysis of GWAS of urticaria from Iceland, the UK and with summary statistics from FinnGen freeze 7 and BioBank Japan for urticaria (total of 40,694 cases and 1,230,001 controls). We have also done a meta-analysis on the same populations for all other tested phenotypes in the study when available.

ii) they should be able to detect some novel associations including *FCER1A* and *STAT6* (they can keep all of the discussion regarding *FCER1A* and *STAT6* since both were significant after meta-analysis as they showed)

As expected, we are now in the meta-analysis of GWAS of urticaria from the four populations detecting novel associations. Common variants at nine loci associate with risk of urticaria. Seven of the associations are novel. The associations represented by the variants at the *FCER1A* and *STAT6* loci reached genome wide significance in the meta-analysis. The additional associations were represented by common variants in *C4A*, *TPSD1*, *ZFPMI*, *CBLB*, *NFKB1*, and *TBLIXR1*. The minor allele of variants at *TPSD1*, *NFKB1*, and *TBLIXR1* associate with increased risk of urticaria (OR 1.04-1.06) and the minor allele of the variants at *C4A*, *ZFPMI*, and *CBLB* associate with decreased risk of urticaria (OR 0.93-0.94). The variants are at genes participating in type 2 immune responses and/or mast cell biology (*CBLB*, *FCER1A*, *GCSAML*, *STAT6*, *TPSD1*, *ZFPMI*), innate immunity (*C4A*), and NF- κ B signalling. As in the initial submission, we assessed the association of all nine variants with RNA expression (amount and splicing) and plasma proteomics (cis and trans pQTLs).

iii) they can test the overlap of urticaria-associated variants with IgE-associated ones.

We tested the association of the nine urticaria variants with IgE levels, which can be seen in Tables S9 & S10. Out of the nine urticaria associated variants, only the variants at *FCER1A* and *STAT6* associate with IgE levels. The analysis using biological priors of IgE to identify associations between sequence variants and urticaria have been removed from the manuscript.

This will produce a more robust discussion. I understand that the two urticaria-associated variants at FCER1A and STAT6 were significant after meta-analyzing their results with other recent GWAS. Therefore, I don't think these associations are false positives. My point is that even if they report the same associations (FCER1A and STAT6), the reasonable process of reaching the conclusion is vital in scientific articles.

We appreciate the reviewer comments. Here above we have answered each of the reviewer's suggestions of changes to the study and we made adjustments to the manuscript according to them. Manuscript adjustments can be seen in track changes in an attached file.

2, The authors need to interpret the intercept very carefully. In Loh et al (PMID:29892013; the article they cited in the response letter), Loh explained the tricky aspect of the intercept in the main text as follows: “we observed that while the value of the LD score regression intercept (previously proposed as an indicator of confounding) was generally difficult to interpret due to attenuation bias, which causes the intercept to rise above 1 with increased sample size and heritability”. So, the paper they cited was claiming that we need to interpret intercepts carefully due to the attenuation bias (see the definition in Loh et al). In GWAS results with a large sample size (such as QTL-GWAS reported in Loh et al), we usually observe increased mean chi-squared, which leads to increased intercept (the intercept value in large-scale GWAS tends to be much larger than those in moderate-power GWAS). Therefore, the intercept values in Loh et al can never be fairly compared with this

GWAS’s intercept. How was the attenuation ratio, which was described in Loh et al. The authors can fairly compare their attenuation ratio with those reported in Loh et al (the attenuation ratio is not influenced by the scale of GWAS). If the attenuation ratio is very high compared with the standard value (as reported in Loh et al), this fact should be discussed in the main text as a potential limitation of this GWAS.

Again, the use of intercept as a correction factor of GWAS is just one remedy for the bias in this GWAS. But, of course, this strategy is not at all perfect. Explaining to the readers the potential bias in the GWAS is crucial in scientific articles.

As suggested by the reviewer we have addressed high attenuation ratio in the manuscript and toned back our interpretation of the attenuation ratio. Lines 354-359, page 12 now read:

„We adjusted the χ^2 statistic from our GWAS scan by dividing it with the intercept (correction factor) from the LD score regression analysis. The correction factors that we obtained are consistent with the ones reported by others. For urticaria they were 1.02 for the UK and 1.10 for Iceland. For urticaria we observed a high attenuation ratio of 0.71 for Iceland and 0.53 for the UK. As pointed out previously (cite PMID:29892013), correction factor from LD score regression may be too high (attenuation bias) which may lead to over adjustment and conservative P-values. “

REVIEWERS' COMMENTS:

Reviewer #2 (Remarks to the Author):

I confirmed that the authors have addressed all my comments, and I am now satisfied with the revised manuscript. I do not have further comments.